# An atypical NLR protein modulates the NRC immune receptor network in *Nicotiana benthamiana*

Hiroaki Adachi[1,2,3]*, Toshiyuki Sakai[1,2], Adeline Harant[1], Hsuan Pai[1], Kodai Honda[2], AmirAli Toghani[1], Jules Claeys[1], Cian Duggan[4], Tolga O. Bozkurt[4], Chih-hang Wu[1,5]*, Sophien Kamoun[1]*

1 The Sainsbury Laboratory, University of East Anglia, Norwich Research Park, Norwich, United Kingdom, 2 Laboratory of Crop Evolution, Graduate School of Agriculture, Kyoto University, Kyoto, Japan, 3 JST-PRESTO, Saitama, Japan, 4 Department of Life Sciences, Imperial College London, London, United Kingdom, 5 Institute of Plant and Microbial Biology, Academia Sinica, Taipei, Taiwan

* adachi.hiroaki.3s@kyoto-u.ac.jp (HA); wuchh@gate.sinica.edu.tw (CHW); sophien.kamoun@tsl.ac.uk (SK)

## Abstract

The NRC immune receptor network has evolved in asterid plants from a pair of linked genes into a genetically dispersed and phylogenetically structured network of sensor and helper NLR (nucleotide-binding domain and leucine-rich repeat-containing) proteins. In some species, such as the model plant *Nicotiana benthamiana* and other Solanaceae, the NRC (NLR-REQUIRED FOR CELL DEATH) network forms up to half of the NLRome, and NRCs are scattered throughout the genome in gene clusters of varying complexities. Here, we describe NRCX, an atypical member of the NRC family that lacks canonical features of these NLR helper proteins, such as a functional N-terminal MADA motif and the capacity to trigger autoimmunity. In contrast to other NRCs, systemic gene silencing of *NRCX* in *N. benthamiana* markedly impairs plant growth resulting in a dwarf phenotype. Remarkably, dwarfism of *NRCX* silenced plants is partially dependent on NRCX paralogs NRC2 and NRC3, but not NRC4. Despite its negative impact on plant growth when silenced systemically, spot gene silencing of *NRCX* in mature *N. benthamiana* leaves doesn't result in visible cell death phenotypes. However, alteration of *NRCX* expression modulates the hypersensitive response mediated by NRC2 and NRC3 in a manner consistent with a negative role for NRCX in the NRC network. We conclude that NRCX is an atypical member of the NRC network that has evolved to contribute to the homeostasis of this genetically unlinked NLR network.

## Author summary

Plants have an effective immune system to fight off diverse pathogens such as fungi, oomycetes, bacteria, viruses, nematodes and insects. In the first layer of their immune system, receptor proteins act to detect pathogens and activate the defense response. Plant genomes encode very large and diverse repertoires of immune receptors, some of which

**Data Availability Statement:** RNA-seq data generated in this study have been deposited under the BioProject accessions(https://www.ncbi.nlm.nih.gov/gds): PRJEB55392 and PRJEB55516. All

relevant data are within the manuscript and its Supporting Information files.

**Funding:** This work was funded by the Gatsby Charitable Foundation (TSL core funding) and Biotechnology and Biological Sciences Research Council (BBS/E/J/000PR9795) awarded to SK. SK also receives funding from the European Research Council (BLASTOFF). HA was funded by the Japan Society for the Promotion of Science (Overseas Research Fellowships, 21K20583 and 22K14893) and Precursory Research for Embryonic Science and Technology (JPMJPR21D1). HA received a salary from Japan Society for the Promotion of Science (Overseas Research Fellowships) and Precursory Research for Embryonic Science and Technology (JPMJPR21D1). More information about the funding sources can be found at the following web addresses: the Gatsby Charitable Foundation (https://www.gatsby.org.uk/), Biotechnology and Biological Sciences Research Council (https://www.ukri.org/councils/bbsrc/), the European Research Council (https://erc.europa.eu), the Japan Society for the Promotion of Science (https://www.jsps.go.jp/english/) and Precursory Research for Embryonic Science and Technology (https://www.jst.go.jp/kisoken/presto/en/index.html). The funders had no role in study design, data collection and analysis, decision to publish, or preparation of the manuscript.

**Competing interests:** I have read the journal's policy and the authors of this manuscript have the following competing interests: S.K. receives funding from industry on NLR biology.

function in pairs or as complex receptor networks. However, the immune system can come at a cost for plants and inappropriate receptor activation results in growth suppression and autoimmunity. Here, we show that an atypical immune receptor gene functions as a modulator of the immune receptor network. This type of receptor gene evolved to maintain homeostasis of the immune system and balance fitness trade-offs between growth and immunity. Further understanding how plants regulate their immune receptor system should help guide breeding disease resistant crops with limited fitness penalties.

## Introduction

Plants are invaded by a multitude of pathogens and pests, some of which threaten food security in recurrent cycles of destructive epidemics. Yet, most plants are resistant to most parasites through their highly effective immune system. Plant defense consists of an expanded and diverse repertoire of immune receptors: cell-surface pattern recognition receptors (PRRs) and intracellular nucleotide-binding domain and leucine-rich repeat-containing (NLRs) proteins [1]. Pathogen-associated molecular patterns (PAMPs) are recognized in the extracellular space by PRRs, resulting in pattern-triggered immunity (PTI) [2,3]. NLRs perceive pathogen secreted proteins known as effectors and induce robust immune responses that generally include hypersensitive cell death [4–6]. NLR-mediated immunity (also known as effector-triggered immunity) can be effective in restricting pathogen infection at invasion sites, and NLRs have also been recently shown to be involved in PRR-mediated signalling [7–10]. However, NLR-mediated immunity comes at a cost for plants. NLR mis-regulation and inappropriate activation can lead to deleterious physiological phenotypes, resulting in growth suppression and autoimmunity [11], and the evolution of the plant immune system is constrained by fitness trade-offs between growth and immunity [12,13]. However, our knowledge of the mechanisms by which diverse plant NLRs are regulated is still somewhat limited. Understanding how plants maintain NLR network homeostasis should help guide breeding disease resistant crops with limited fitness penalties.

NLRs occur across all kingdoms of life and generally function in innate immunity through "non-self" perception of invading pathogens [4,14,15]. Plant NLRs share a multidomain architecture typically consisting of a central NB-ARC (nucleotide-binding domain shared with APAF-1, various R-proteins and CED-4) and a C-terminal leucine-rich repeat (LRR) domain [16]. Plant NLRs can be sorted into sub-classes based on NB-ARC phylogenetic clustering and the type of N-terminal domain they carry [16,17]. The largest class of NLRs are the CC-NLRs with the Rx-type coiled-coil (CC) domain preceding the NB-ARC domain [16,18,19]. A prototypical ancient CC-NLR is the HOPZ-ACTIVATED RESISTANCE1 (ZAR1), which has remained relatively conserved throughout evolution over tens of millions of years [20,21]. However, the majority of CC-NLRs have massively expanded throughout their evolution, acquiring new activities through sequence diversification and integration of extraneous domains, as well as losing particular molecular features following sub-functionalization [22–24].

Even though some plant NLRs function as singletons carrying both pathogen sensing and immune signalling activities, other NLRs form genetic and biochemical networks with varying degrees of complexity [25–27]. NLRs can also cause deleterious genetic interactions known as hybrid incompatibility, presumably because mismatched NLRs inadvertently activate immunity [12,13]. Hybrid autoimmunity is probably a trade-off of the rapidly evolving NLRome, which is expanding and diversifying in most angiosperm taxa, even at the intraspecific level

[23,28,29]. NLRs can also cause a spontaneous autoimmune phenotype known as lesion mimicry [30]. Classic examples include mutants of the Toll/interleukin-1 receptor (TIR)-NLRs, *ssi4* (*suppressor of SA insensitivity 4*), *snc1* (*suppressor of npr1-1, constitutive 1*), *slh1* (*sensitive to low humidity 1*), *chs1* (*chilling-sensitive mutant 1*), *chs2* and *chs3*, which exhibit autoimmune phenotypes in Arabidopsis [31–37]. Mutations in other Arabidopsis NLR genes, such as *laz5* (*lazarus 5*), *adr1* (*activated disease resistance 1*), *summ2* (*suppressor of mkk1 mkk2, 2*), *rps4* (*resistance to Pseudomonas syringae 4*), *csa1* (*constitutive shade-avoidance 1*), *soc3* (*suppressor of chs1-2, 3*), and *sikic* (*sidekick snc1*) are genetic suppressors of autoimmunity or cell death phenotypes [38–46]. Nonetheless, to date, NLRs are not classed among so-called "lethal-phenotype genes" that are essential for plant viability and survival [47]. This is despite the fact that ~500 *NLR* genes have been experimentally studied [16].

Our understanding of molecular mechanisms underpinning plant NLR activation and the subsequent signalling events has significantly advanced with the elucidation of NLR protein structures. Activated CC-NLR ZAR1, TIR-NLRs RECOGNITION OF XOPQ 1 (ROQ1) and RESISTANCE TO PERONOSPORA PARASITICA 1 (RPP1) oligomerize into multimeric complexes known as resistosomes [48–51]. In the case of ZAR1, activation by pathogens induces a switch from ADP to dATP/ATP binding and oligomerization into a pentameric resistosome [49]. This results in a conformational 'death switch', with the five N-terminal α1 helices forming a funnel-shaped structure that acts as a $Ca^{2+}$ channel on the plasma membrane [52]. The α1 helix of ZAR1 and about one-fifth of angiosperm CC-NLRs are defined by a molecular signature, the MADA motif [24]. This α1/MADA helix is interchangeable between distantly related NLRs indicating that the 'death switch' mechanism applies to MADA-CC-NLRs from diverse plant taxa [24].

One class of MADA-CC-NLRs are the NRCs (NLR-REQUIRED FOR CELL DEATH), that are central nodes in a large NLR immune network of asterid plants [53]. NRCs function as helper NLRs (NRC-H), required for a large number of sensor NLRs (NRC-S) to induce the hypersensitive response and immunity [53]. These NRC-S are encoded by classical disease resistance genes that detect pathogens as diverse as viruses, bacteria, oomycetes, nematodes and insects. NRC-H and NRC-S form phylogenetic sister clades within a wider NRC superclade that makes up to half of the NLRome in the Solanaceae. This NRC superclade emerged early in asterid evolution about 100 million years ago from an ancestral pair of genetically linked NLRs [53]. However, in sharp contrast to paired NLR pairs, functionally connected NRC-H and NRC-S genes are not always clustered and can be scattered throughout the plant genomes [53]. Our current model is that the NRCs and their sensor evolved from bi-functional NLRs that have sub-functionalized and specialized throughout evolution [24,26,53]. In support of this model, the MADA sequence has degenerated in NRC-S in contrast to the NRC-H, which carry functional N-terminal α1 helices [24].

Plant-pathogen coevolution has driven NLRs to form immune receptor networks [25,27]. The emerging paradigm in the field of plant immunity is that helper NLRs, NRCs, as well as the RESISTANCE TO POWDERY MILDEW 8 (RPW8)-type CC-NLRs (CC$_R$-NLRs) ADR1 and N REQUIREMENT GENE 1 (NRG1), form receptor networks with multiple sensor NLRs [54]. These genetically dispersed NLR networks are likely to cause a heightened risk of autoimmunity during plant growth and development. Yet, our knowledge of the regulatory mechanisms that attenuate such deleterious effects of NLR networks, especially Solanaceae NRC networks, are limited. Here, we describe NRCX, an atypical NLR protein that belongs to the NRC-H phylogenetic clade. Gene silencing of *NRCX* markedly impairs plant growth, presumably because of mis-activation of its helper NLR paralogs NRC2 and NRC3. We propose that NRCX maintains NRC network homeostasis by modulating the activities of key helper NLR nodes during plant growth.

## Results

### Systemic silencing of *NRCX* impairs *Nicotiana benthamiana* growth

While performing virus-induced gene silencing (VIGS) of NRC family genes in the model plant *Nicotiana benthamiana* (S1 Fig), we found that silencing of NLR NbS00030243g0001.1

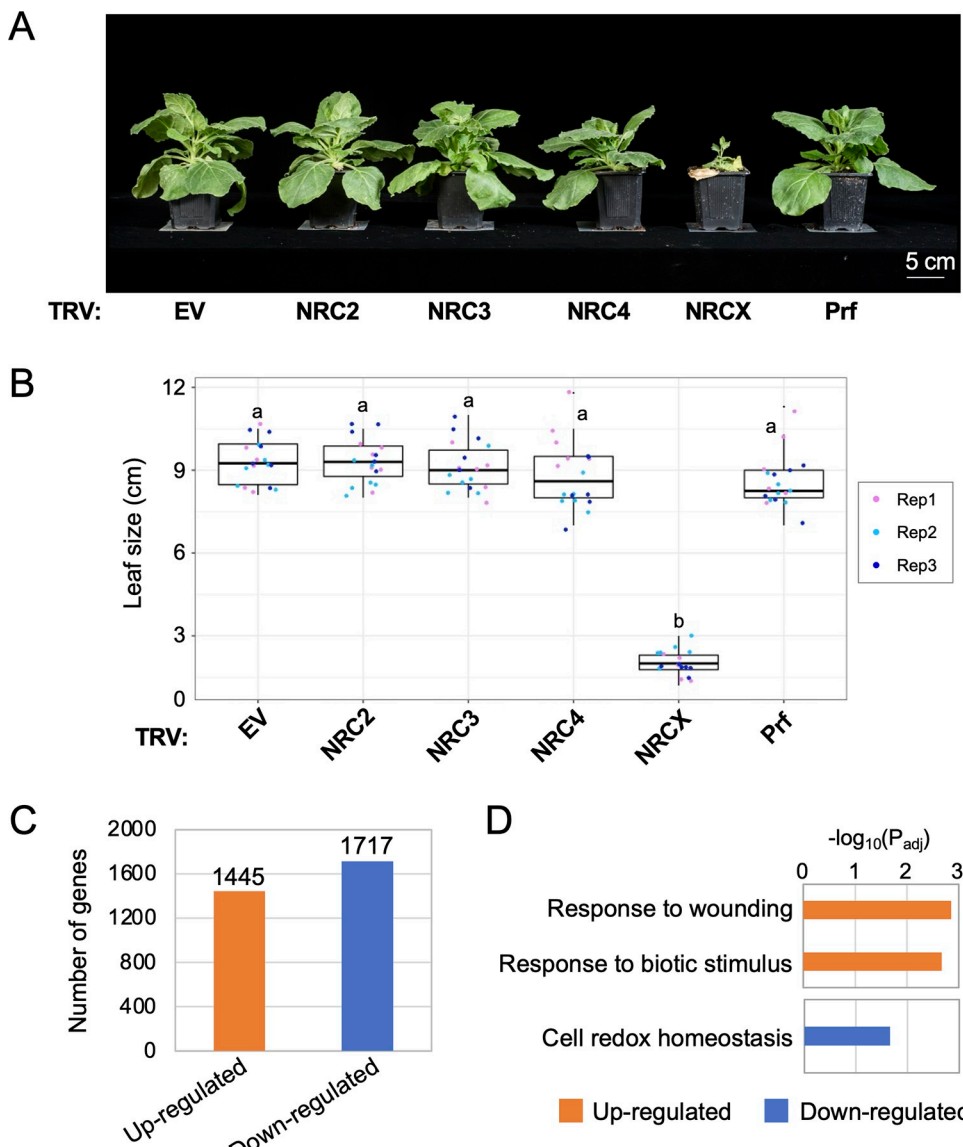

**Fig 1. Virus-induced gene silencing of *NRCX* impairs *Nicotiana benthamiana* growth. (A)** The morphology of 6-week-old *NRC2-*, *NRC3-*, *NRC4-*, *NRCX-* or *Prf*-silenced *N. benthamiana* plants. 2-week-old *N. benthamiana* plants were infiltrated with *Agrobacterium* strains carrying tobacco rattle virus (TRV) VIGS constructs, and photographs were taken 4 weeks after the agroinfiltration. TRV empty vector (TRV:EV) was used as a negative control. **(B)** Quantification of the leaf size of 6-week-old *NRC2-*, *NRC3-*, *NRC4-*, *NRCX-* or *Prf*-silenced *N. benthamiana* plants. One leaf per plant was harvested from the same position (the 5th leaf from cotyledons) and was used for measuring the leaf diameter. Data was obtained from 18 different VIGS plants in three independent experiments. Statistical differences among the samples were analysed with Tukey's HSD test (p<0.01). **(C)** The number of up-regulated genes ($\log_2$(TRV:*NRCX*/TRV:*GUS*) ≧ 1 and P-value ≦ 0.05) and down-regulated genes ($\log_2$(TRV:*NRCX*/TRV:*GUS*) ≦ -1 and P-value ≦ 0.05) in TRV:*NRCX* leaf tissue compared to TRV:*GUS* control. **(D)** Enriched GO terms in up-regulated and down-regulated genes identified in C.

(referred to from here on as NRCX) causes a severe dwarf phenotype (Fig 1A and 1B). This VIGS construct was made by using 5' 446-bp sequence of *NRCX* gene that does not hit off-target candidate genes in Solanaceae Genomics Network VIGS Tool search (https://vigs.solgenomics.net/) using Nicotiana benthamiana v0.4.4 and Nicotiana benthamiana v1.0.1 databases (S2 Fig). In these VIGS experiments, we also independently targeted NRC helper (NRC-H) genes, *NRC2*, *NRC3* and *NRC4*, as well as the NRC sensor (NRC-S) gene *Prf* (NRC2/3 dependent) for silencing in *N. benthamiana* [53,55]. Yet, none of the NRC-H and NRC-S silenced *N. benthamiana* plants showed quantitative growth defects (Fig 1A and 1B). These results suggest that *NRCX* is unique among NLR genes described to date as a "lethal-phenotype" gene according to the definition of Lloyd et al. [47].

To determine whether constitutive defense activation occurs in the dwarf plants of TRV:*NRCX*, total RNA was extracted from leaf tissue of four-week-old TRV:*NRCX* and TRV:*GUS* plants and were subjected to RNA-seq analysis. In total, 3162 genes were detected as differentially expressed genes (DEGs, TRV:*NRCX* vs. TRV:*GUS*), in which 1445 and 1717 genes were up-regulated and down-regulated, respectively (Fig 1C, S1 and S2 Files). The DEGs do not include NRC-H genes such as *NRC2*, *NRC3* and *NRC4* (S1 and S2 Files). Gene Ontology (GO) analysis showed significant enrichment of GO terms 'response to wounding' and 'response to biotic stimulus' in up-regulated DEGs, while 'cell redox homeostasis' is enriched in down-regulated DEGs (Fig 1D). DEGs having the GO terms 'response to wounding' and 'response to biotic stimulus' include genes encoding Proteinase inhibitor I and pathogenesis related proteins (S3 File). These results suggest that constitutive defense response is induced by *NRCX* systemic silencing, thereby resulting in the dwarf phenotype in *N. benthamiana* plants.

## NRCX is a CC-NLR in the NRC helper phylogenetic clade

We investigated the precise phylogenetic position of NRCX in the NRC superclade. First, we built a phylogenetic tree with 431 CC-NLRs, including the CC-NLRome from four representative plant species (Arabidopsis, sugar beet, tomato and *N. benthamiana*) and 16 representative CC-NLRs (Fig 2A). Next, we extracted the NRC-H subclade NLRs, which includes NRCX, for further phylogenetic analysis (Fig 2A and 2B). In this NRC-H subclade, NRCX forms a small clade together with a tomato NLR (Solyc03g005660.3.1; named as SlNRCX) (Figs 2B and S3). The NRCX clade is more closely related to NRC1/2/3 subclade than to the NRC4 subclade (Fig 2B).

We further searched *NRCX* ortholog from other plant species by phylogenetic analysis using an NLR dataset including 6408 NLR candidates from nine Solanaceae species (*N. benthamiana*, *Capsicum annuum*, *Capsicum chinense*, *Capsicum baccatum*, *Solanum commersonii*, *Solanum tuberosum*, *Solanum lycopersicum*, *Solanum pennellii* and *Solanum chilense*) and three Convolvulaceae species (*Ipomoea triloba*, *Ipomoea nil* and *Ipomoea trifida*) (S4 File). In total, we identified 5 *NRCX* orthologs from 5 species (*N. benthamiana*, *S. commersonii*, *S. lycopersicum*, *S. pennellii* and *S. chilense*) (S4 and S5 Figs). In the Convolvulaceae species, we did not find *NRC2*, *NRC3*, *NRC4* and *NRCX* ortholog genes (S5 Fig). We also didn't detect *NRCX* ortholog from four Solanaceae reference genome databases (*C. annuum*, *C. chinense*, *C. baccatum* and *S. tuberosum*). We conclude that *NRCX* gene has a patchy distribution across the Solanaceae and co-exists with *NRC2*, *NRC3* and *NRC4* genes in at least five Solanaceae plant species (S5 Fig).

We scanned NRCX and other NRC-H proteins for conserved sequence motifs (Fig 2C). NRC-H members typically have the N-terminal MADA motif that is functionally conserved across many dicot and monocot CC-NLRs and is required for hypersensitive cell death and disease resistance [24]. We ran the HMMER software [56] to query NRCX with a previously

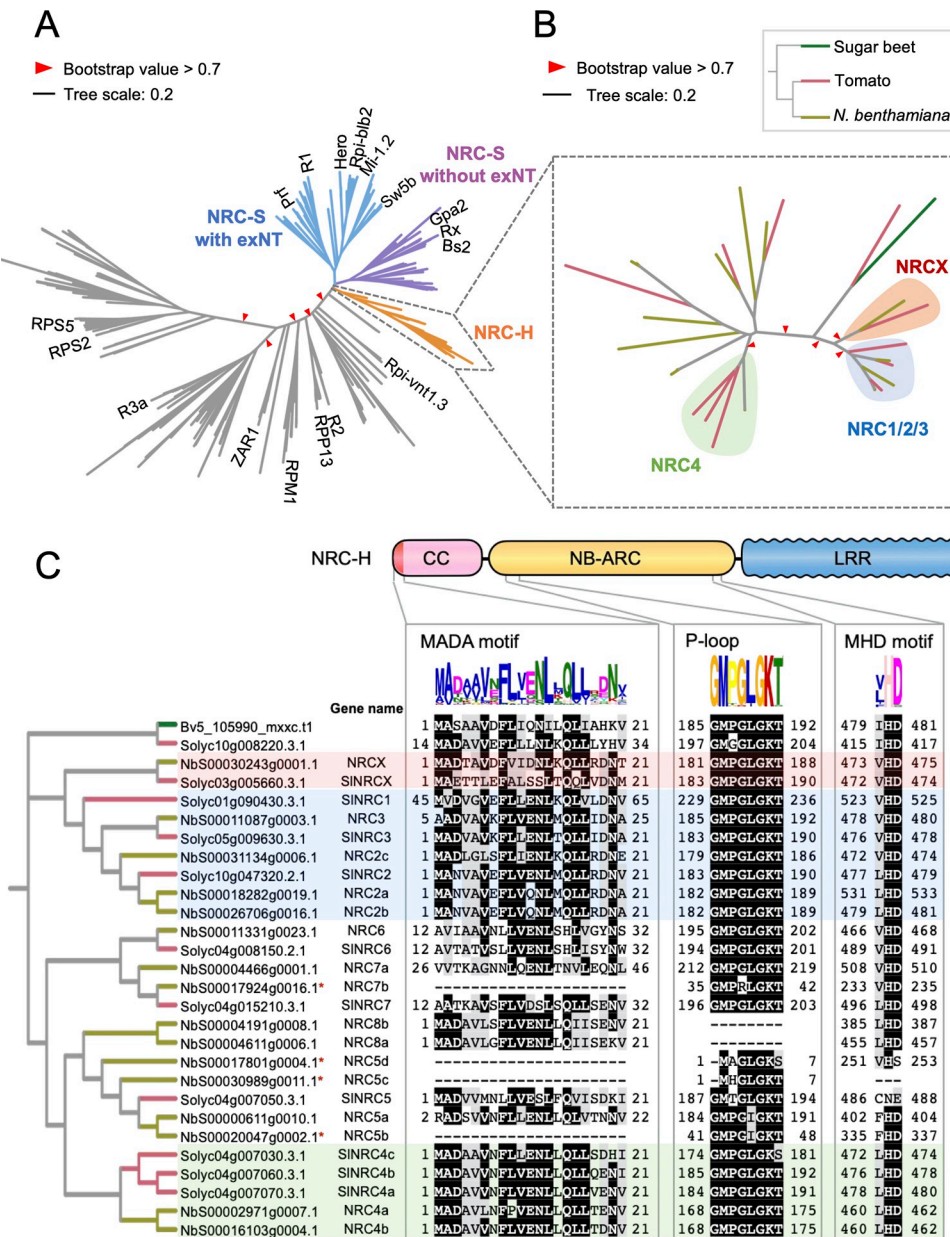

**Fig 2. NRCX is a CC-NLR in the NRC-helper clade. (A, B)** NRCX is an NRC-helper (NRC-H) member phylogenetically closely related to the NRC1/2/3 clade. The maximum likelihood phylogenetic tree was generated in RAxML version 8.2.12 with JTT model using NB-ARC domain sequences of 431 NLRs identified from *N. benthamiana* (NbS-), tomato (Solyc-), sugar beet (Bv-), and Arabidopsis (AT-) (S7 File). The NRC superclade containing NRC-H and NRC-sensor (NRC-S) clades are described with different branch colours. The NRC-S clade is divided into NLRs that lack an extended N-terminal domain (exNT) prior to their CC domain and those that carry an exNT. The NRC-H clade phylogenetic tree is shown with different colours based on plant species (B). Red arrow heads indicate bootstrap support > 0.7 and is shown for the relevant nodes. The scale bars indicate the evolutionary distance in amino acid substitution per site. **(C)** Domain and motif architectures of NRC-H clade members. Amino acid sequences of MADA motif, P-loop and MHD motif are mapped onto the NRC-H phylogenetic tree. Each motif was identified in MEME using NRC-H sequences. NRCX, NRC1/2/3 and NRC4 clades are highlighted in red, blue and green, respectively. Red asterisks on gene name describe truncated genes at their N terminus.

reported MADA motif-Hidden Markov Model (HMM) [24]. This HMMER search predicted a MADA sequence at the N-terminus of NRCX (HMM score = 22.2) and in all other NRC-H except in four *N. benthamiana* NRC-H proteins that have N-terminal truncations (NbS00017924g0016.1, NbS00017801g0004.1, NbS00030989g0011.1 and NbS00020047g0002.1) (Fig 2C). In addition, NRCX carries intact P-loop and MHD motif in its NB-ARC domain like the majority of CC-NLRs (Fig 2C). We noted that the P-loop wasn't predicted in NbS00004191g0008.1 and NbS00004611g0006.1, and the MHD motif is absent or divergent in NbS00030989g0011.1, NbS00017801g0004.1 and Solyc04g007050.3.1 (Fig 2C). Taken together, we conclude that NRCX has the typical sequence motifs of MADA-CC-NLRs, similar to the great majority of proteins in the NRC-H subclade.

## Unlike other NRC helpers, mutations in the MHD motif fail to confer autoactivity to NRCX

Even though NRCX carries canonical features of MADA-CC-NLRs, we investigated whether it can execute the hypersensitive cell death similar to other NRC helper NLRs. Histidine (H) to arginine (R) or aspartic acid (D) to valine (V) substitutions within the MHD motif confer autoactivity to the NRC helpers NRC2, NRC3 and NRC4 [57]. To test if NRCX has cell death induing activity like other helper NRCs, we performed site-directed mutagenesis of the MHD motif predicted in the NB-ARC domain of NRCX. We first introduced the H474R and D475V mutations in NRCX (referred to as NRCX^HR and NRCX^DV, respectively) (Fig 3A). Both NRCX^HR and NRCX^DV did not induce macroscopic cell death when expressed in *N. benthamiana* leaves using agroinfiltration, in contrast to an NRC4 autoactive MHD motif mutant (NRC4^DV) (Fig 3B). To further challenge this finding, we randomly mutated NRCX H474 and D475 residues in the MHD motif and used agroinfiltration to express them in *N. benthamiana* leaves (S6 Fig). None of the 55 independent NRCX MHD mutants we tested induced visible macroscopic cell death (S6 Fig). These results indicate that NRCX does not have the capacity to induce cell death, unlike the other NRC-H it is related to.

## The N-terminal MADA motif of NRCX is not functional and doesn't mediate hypersensitive cell death

To further investigate why NRCX cannot execute the cell death activity, we determined whether or not the MADA motif of NRCX is functional using the motif swap strategy we previously developed [24]. To this end, we swapped the first 17 amino acids of the autoactive NRC4^DV with the equivalent region of NRCX, resulting in a MADA^NRCX-NRC4^DV chimeric protein (Fig 4A and 4B). The N-terminal sequence of NRC4 can be functionally replaced with matching sequences of other MADA-CC-NLRs, including NRC2 from *N. benthamiana*, ZAR1 from Arabidopsis and even the monocot MADA-CC-NLRs MLA10 and Pik-2 [24]. Intriguingly, MADA^NRCX-NRC4^DV did not induce any visible cell death when expressed by agroinfiltration in *N. benthamiana* leaves unlike the positive controls MADA^NRC2-NRC4^DV and MADA^ZAR1-NRC4^DV (Fig 4A–4C). MADA^NRCX-NRC4^DV chimeric protein accumulated to similar levels as MADA^ZAR1-NRC4^DV in *N. benthamiana* leaves, indicating that the lack of activity was probably not due to protein destabilization (Fig 4E). These results indicate that the MADA region of NRCX does not have the capacity to trigger hypersensitive cell death in the NRC4 protein background, and therefore despite its sequence conservation, it probably forms a non-functional N-terminal α1 helix.

We also performed the opposite motif swap to test whether functional MADA sequences confer gain of cell death activity to NRCX. To this end, we produced MADA chimera constructs in the NRCX^DV and NRCX^HR MHD motif mutant backgrounds (S7A Fig). Although

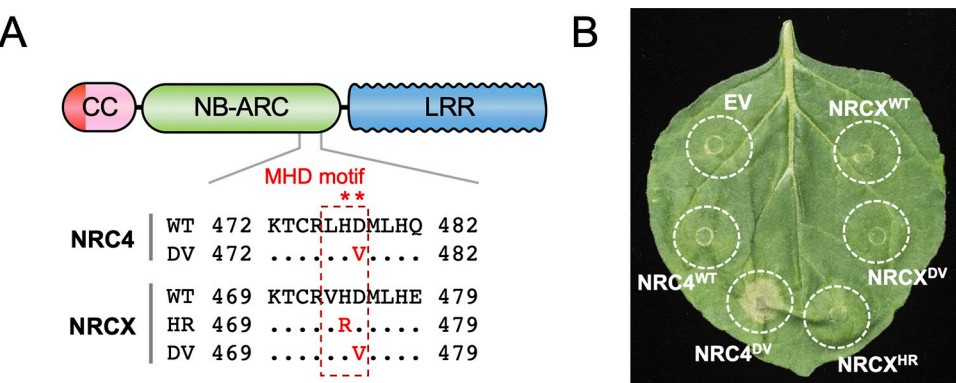

**Fig 3. Mutations in the NRCX MHD motif do not result in autoactive cell death in *Nicotiana benthamiana*. (A)** Schematic representation of the mutated sites in the NRC4 and NRCX MHD motifs. Substituted residues are shown in red in the multiple sequence alignment. **(B)** NRC4^WT, NRCX^WT and the MHD mutants were expressed in *N. benthamiana* leaves by agroinfiltration. Cell death phenotype induced by the MHD mutant was recorded 5 days after the agroinfiltration. Quantification of the cell death intensity is shown in S6 Fig.

the N-terminal MADA sequences from NRC2 and ZAR1 are functional in NRC4^DV (Fig 4) [24], MADA^NRC2-NRCX^DV, MADA^ZAR1-NRCX^DV, MADA^NRC2-NRCX^HR and MADA^ZAR1-NRCX^HR did not induce any visible cell death when expressed in *N. benthamiana* leaves (S7B and S7C Fig). We further tested NRCX^DV and NRCX^HR chimera constructs with NRC4 MADA sequence (S7A Fig). Neither MADA^NRC4-NRCX^DV nor MADA^NRC4-NRCX^HR caused visible cell death response in *N. benthamiana* leaves (S7B and S7C Fig). These results suggest that in addition to the non-functional MADA region, there is other constrain(s) in NRCX protein that led to the incapacity to cause autoactive cell death unlike other helper NRCs.

## The dwarf phenotype of *NRCX*-silenced *Nicotiana benthamiana* plants is partially dependent on NRC2 and NRC3 but not NRC4

We hypothesized that NRCX has a regulatory role in the NRC network, given that it belongs to the NRC helper clade but lacks the capacity to cause hypersensitive cell death. To test this hypothesis, we silenced *NRCX* in three different *N. benthamiana* lines, *nrc2/3*, *nrc4* and *nrc2/3/4*, that we previously described as carrying loss-of-function mutations in NRC2, NRC3 and/ or NRC4 [24,58,59] (Fig 5). Four weeks after inoculation with TRV:*NRCX*, we observed partial suppression of the TRV:*NRCX*-mediated growth defects in *nrc2/3* and *nrc2/3/4* plants, but not in *nrc4* plants (Fig 5A–5C). We confirmed that the quantitative differences were reproducible by using at least two independent lines of each of the three mutants (Fig 5A–5C). In these experiments, we did not observe quantitative growth differences between *nrc2/3* and *nrc2/3/4* plants (Fig 5B and 5C).

To independently challenge these results, we performed co-silencing experiments where *NRCX* was knocked-down together with *NRC2/3*, *NRC4* or *NRC2/3/4*. To silence multiple genes by VIGS in *N. benthamiana* plants, we tandemly cloned gene fragments of *NRCX* and other NRCs in the same TRV expression vector. Compared to TRV:*NRCX* plants, the dwarf phenotype was partially suppressed in TRV:*NRC2/3/X* and TRV:*NRC2/3/4/X*, but not in TRV: *NRC4/X* plants (S8A–S8C Fig). Each silencing construct specifically reduced mRNA levels of the target *NRC* gene (S8D Fig), indicating that phenotypic differences are unlikely to be due to off-target gene silencing effects. Altogether, we conclude that the TRV:*NRCX*-mediated dwarf phenotype is partially dependent on NRC2 and NRC3, but not NRC4.

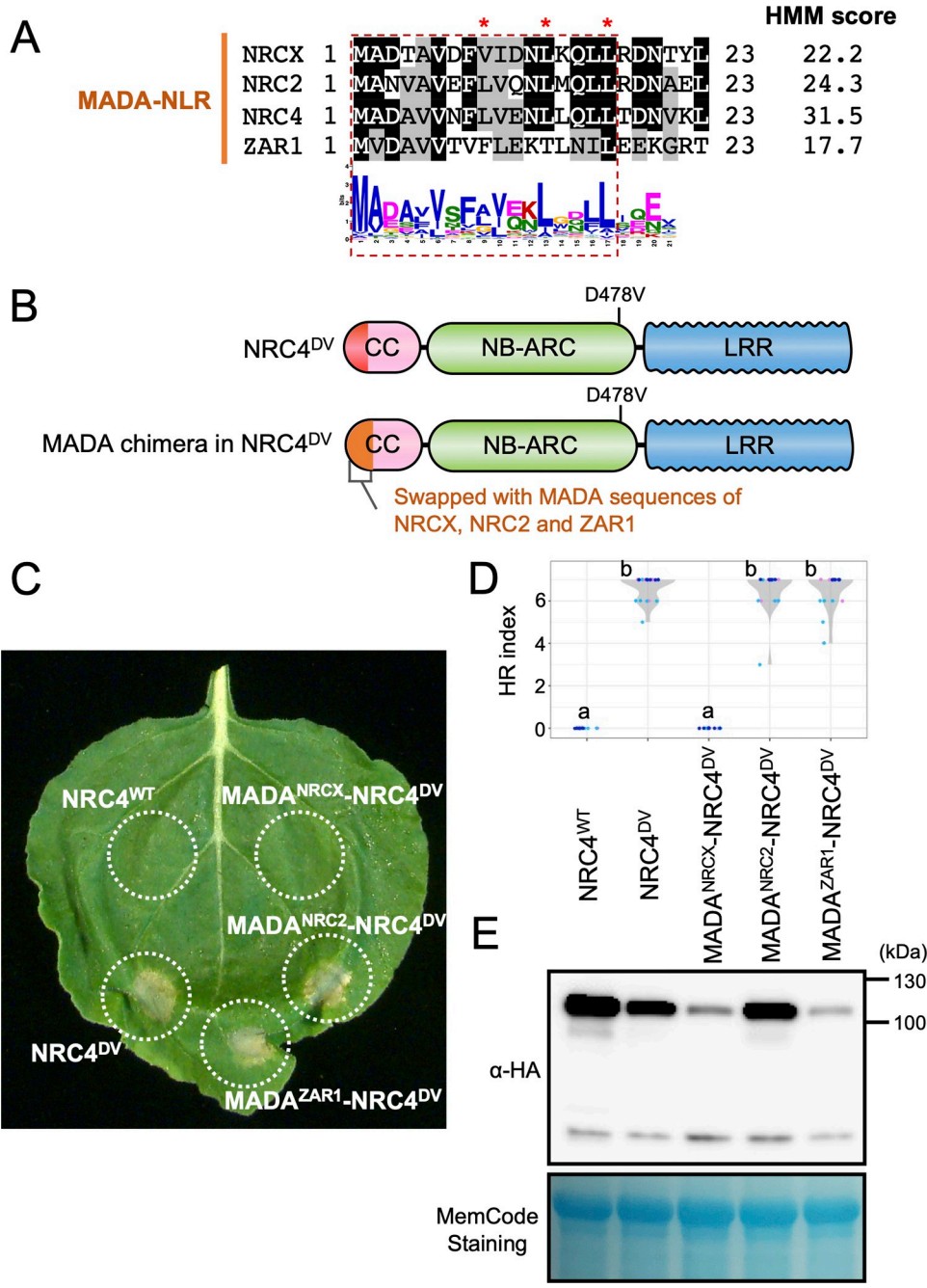

**Fig 4. Unlike other MADA-CC-NLRs, the N-terminal 17 amino acids of NRCX fails to confer cell death activity to an NRC4 autoactive mutant.** (A) Alignment of the N-terminal region of the MADA-CC-NLRs, NRCX, NRC2, NRC4 and ZAR1. Key residues for cell death activity [24] are marked with red asterisks in the sequence alignment. Each HMM score is indicated. (B) Schematic representation of NRC4 MADA motif chimeras. The first 17 amino acid region of NRCX, NRC2 and ZAR1 was swapped into the autoactive NRC4 mutant (NRC4DV), resulting in the NRC4 chimeras with MADA sequences originated from other MADA-CC-NLRs. (C) Cell death phenotypes induced by the NRC4 chimeras. NRC4WT-6xHA, NRC4DV-6xHA and the chimeras were expressed in *N. benthamiana* leaves by agroinfiltration. Photographs were taken at 5 days after the agroinfiltration. (D) Violin plots showing cell death intensity scored as an HR index. Data was obtained from 18 different replicates in three independent experiments. Statistical differences among the samples were analyzed with Tukey's honest significance difference (HSD) test (p<0.01). (E) *In planta* accumulation of the NRC4 variants. For anti-HA immunoblots of NRC4 and the mutant proteins, total proteins were prepared from *N. benthamiana* leaves at 1 day after the agroinfiltration. Equal loading was checked with Reversible Protein Stain Kit (Thermo Fisher).

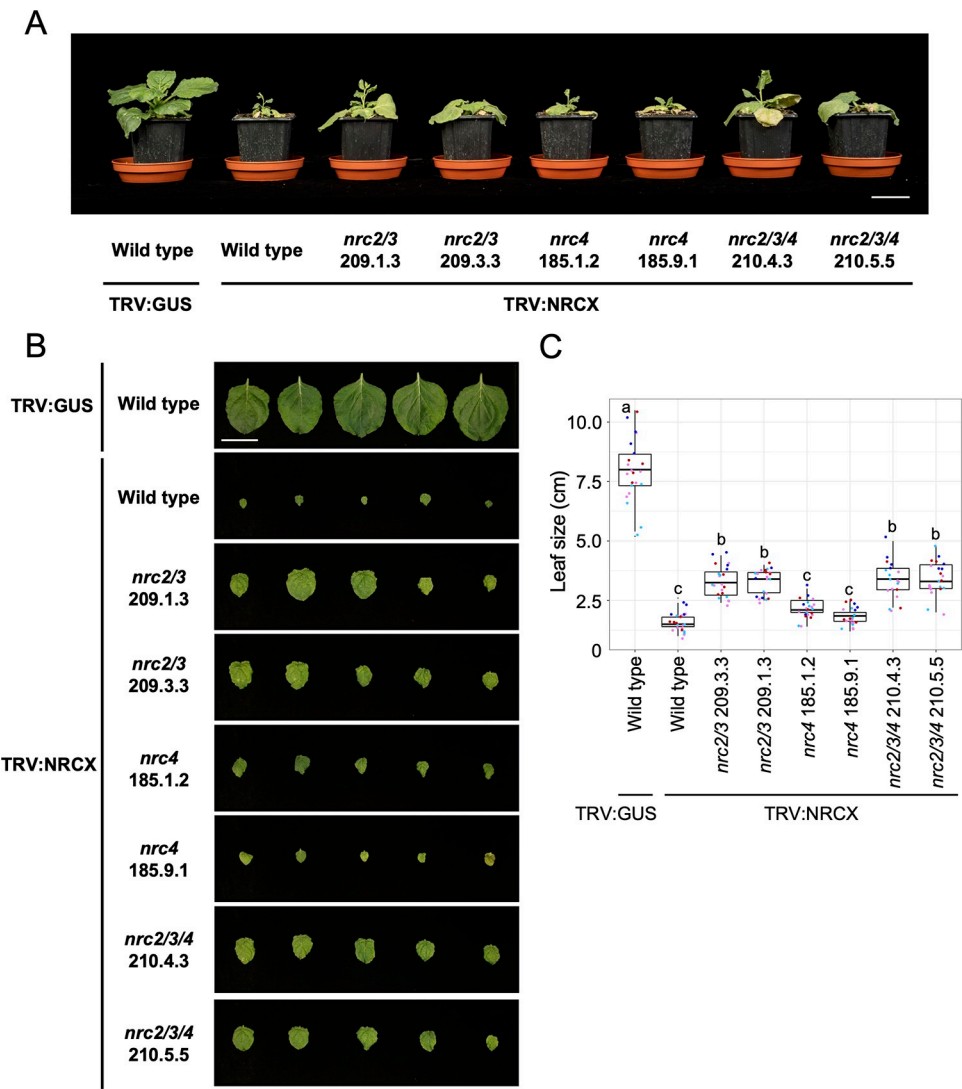

**Fig 5. The dwarf phenotype by *NRCX*-silenced *Nicotiana benthamiana* plants is partially dependent on NRC2 and NRC3 but not NRC4. (A)** The morphology of 6-week-old wild-type *N. benthamiana*, *nrc2/3*, *nrc4* and *nrc2/3/4* CRISPR-knockout lines expressing TRV:*NRCX*. 2-week-old wild-type and the knockout plants were infiltrated with *Agrobacterium* strains carrying TRV:*GUS* or TRV:*NRCX*, and photographs were taken 4 weeks after the agroinfiltration. **(B, C)** Quantification of the leaf size. One leaf per each plant was harvested from the same position (the 5th leaf from cotyledons) and was used for measuring the leaf diameter. Data was obtained from 20 to 22 different VIGS plants in four independent experiments. Statistical differences among the samples were analyzed with Tukey's HSD test (p<0.01). Scale bars = 5 cm.

## Hairpin RNA-mediated gene silencing of *NRCX* in *Nicotiana benthamiana* leaves doesn't result in visible cell death phenotypes

To gain further insights into NRCX activities, we generated a hairpin RNA (hpRNA) silencing construct (hpRNA:*NRCX*) for *NRCX* silencing after transient expression in mature *N. benthamiana* leaves. First, we investigated the degree to which the hpRNA:*NRCX* expression causes cell death in *N. benthamiana* leaves, given that NLR-mediated dwarfism in plants is often linked to cell death [40]. Five days after the agroinfiltration, hpRNA-mediated gene silencing of *NRCX* did not result in macroscopic cell death in *N. benthamiana* leaves whereas

co-expression of Pto/AvrPto resulted in a visible cell death response (S9A Fig). To monitor cell death at a microscopic level, we performed trypan blue staining, which generally visualizes dead cells. Trypan blue only stained trichomes in the hpRNA:*NRCX* and hpRNA:*GUS* (negative control) treated leaf panels, whereas it clearly revealed cell death in epidermal and mesophyll cells in the Pto/AvrPto treatment (positive control) (S9B and S9C Fig). These observations indicate that hpRNA-mediated gene silencing of *NRCX* does not cause visible cell death in *N. benthamiana* leaves. This hpRNA-mediated gene silencing strategy also enabled us to bypass the severe dwarf phenotype observed in whole-plant VIGS experiments to perform functional analyses of *NRCX*.

## Hairpin RNA-mediated gene silencing of *NRCX* enhances NRC2- and NRC3-dependent cell death

Our finding that mutations in *NRC2* and *NRC3* are genetic suppressors of TRV:*NRCX*-mediated dwarfism raises the possibility that NRCX negatively modulates the activity of these helper NLRs and prompted us to investigate this hypothesis. To this end, we co-expressed hpRNA:*NRCX* by agroinfiltration in *N. benthamiana* leaves with NRC-dependent sensor NLRs (Pto/SlPrf, Gpa2, Rpi-blb2, R1, Sw-5b and Rx) or NRC-independent NLRs (R2 and R3a) and their matching pathogen effectors [53,57]. hpRNA-mediated gene silencing *NRCX* enhanced the hypersensitive cell death triggered by the sensor NLRs SlPrf, Gpa2 (NRC2 and NRC3 dependent) and Sw-5b (NRC2, NRC3 and NRC4 dependent) relative to the hpRNA:*GUS* control, but did not affect the other sensor NLRs, including Rpi-blb2, R1, Rx, R2 and R3a (Figs 6A, 6B and S10). It should be pointed out that although NRC2, NRC3 and NRC4 redundantly contribute to effector-activated Sw-5b hypersensitive cell death [53], we recently showed that an autoactive mutant of Sw-5b (Sw-5b$^{D857V}$) signals only through NRC2 and NRC3 [57]. Therefore, we conclude that knocking-down of *NRCX* enhances the activities of NRC2- and NRC3-dependent NRC-S, but doesn't affect the activity of NRC-S that are only dependent on NRC4, as well as other NLR outside the NRC network.

Next, we investigated the extent to which NRCX affects the activities of autoactive mutants of the NRC-H, NRC2$^{H480R}$ (NRC2$^{HR}$), NRC3$^{D480V}$ (NRC3$^{DV}$) and NRC4$^{D478V}$ (NRC4$^{DV}$), which cause cell death even in the absence of pathogen effectors [57]. We co-expressed NRC2$^{HR}$, NRC3$^{DV}$ and NRC4$^{DV}$ with hpRNA:*NRCX* or hpRNA:*GUS* by agroinfiltration in *N. benthamiana* leaves. *NRCX* silencing enhanced the cell death responses caused by NRC2$^{HR}$ and NRC3$^{DV}$, but not NRC4$^{DV}$, compared to the hpRNA:*GUS* silencing control (Fig 6C, 6D and S11). hpRNA:*NRCX* expression reduced mRNA levels of endogenous *NRCX* gene, but did not significantly alter the expression of other NRCs, indicating that the enhanced cell death phenotype was probably not due to off-target silencing (Fig 6E). Altogether, these two sets of hpRNA-mediated gene silencing experiments indicate that NRCX modulates the helper NLRs NRC2 and NRC3 nodes in the NRC network.

## NRCX overexpression compromises NRC2 and NRC3, but not NRC4, autoimmune cell death

We further challenged our model by testing the extent to which NRCX overexpression can suppress the cell death caused by autoactive NRC2 and NRC3. We generated an overexpression construct of wild-type NRCX and co-expressed it by agroinfiltration with autoactive NRC2$^{HR}$, NRC3$^{DV}$ and NRC4$^{DV}$ mutants in *N. benthamiana* leaves. Five days after agroinfiltration, we observed that wild-type NRCX expression compromised autoimmune cell death triggered by NRC2$^{HR}$ and NRC3$^{DV}$, but not NRC4$^{DV}$, relative to the empty vector control

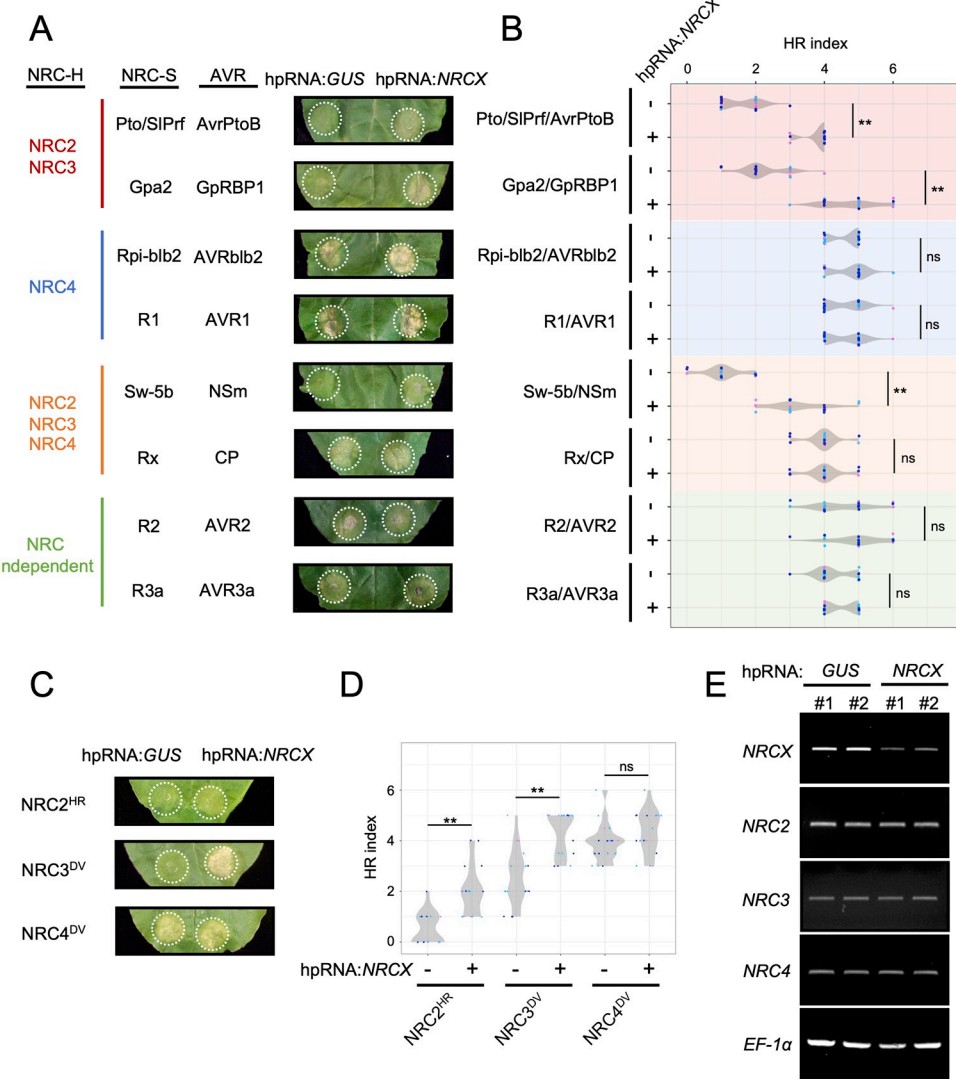

**Fig 6. Hairpin RNA-mediated gene silencing of *NRCX* enhances NRC2- and NRC3-dependent hypersensitive cell death.** (A) Hypersensitive cell death phenotypes after co-expressing different NRC-S and AVR combinations with hpRNA:*GUS* (control) or hpRNA:*NRCX* by agroinfiltration. Cell death intensity was scored at 2–5 days after the agroinfiltration, and photographs were taken at 5 days after the agroinfiltration. (B) Violin plots showing cell death intensity scored as an HR index at 5 days after the agroinfiltration. Time-lapse HR index is shown in S10 Fig. The HR index plots are based on 22 different replicates in three independent experiments. Asterisks indicate statistically significant differences with *t* test (**p<0.01). (C) Autoactive cell death phenotypes induced by MHD mutants of NRC2, NRC3 and NRC4 with hpRNA:*GUS* or hpRNA:*NRCX*. Photos indicate cell death response at 4 days after the agroinfiltration. (D) Violin plots showing cell death intensity scored at 4 days after the agroinfiltration. Time-lapse HR index is shown in S11 Fig. Data was obtained from 18 different replicates in three independent experiments. (E) *NRCX* silencing in *N. benthamiana*. Leaf samples were collected 2 days after agroinfiltration expressing hpRNA:*GUS* and hpRNA:*NRCX*. Total RNA was extracted from two independent plant samples (#1 and #2). The expression of *NRCX* and other *NRC-H* genes were analysed in semi-quantitative RT-PCR using specific primer sets. *Elongation factor 1α* (*EF-1α*) was used as an internal control.

(Fig 7). Taken together, manipulation of *NRCX* expression levels in *N. benthamiana* leaves suggests a negative role of NRCX in NRC2 and NRC3, but not NRC4, mediated immunity.

To further test whether tomato NRCX ortholog has a similar negative role in NRC3 autoactive cell death, we cloned wild-type *SlNRCX* and overexpressed it with NRC3^DV mutant in *N*.

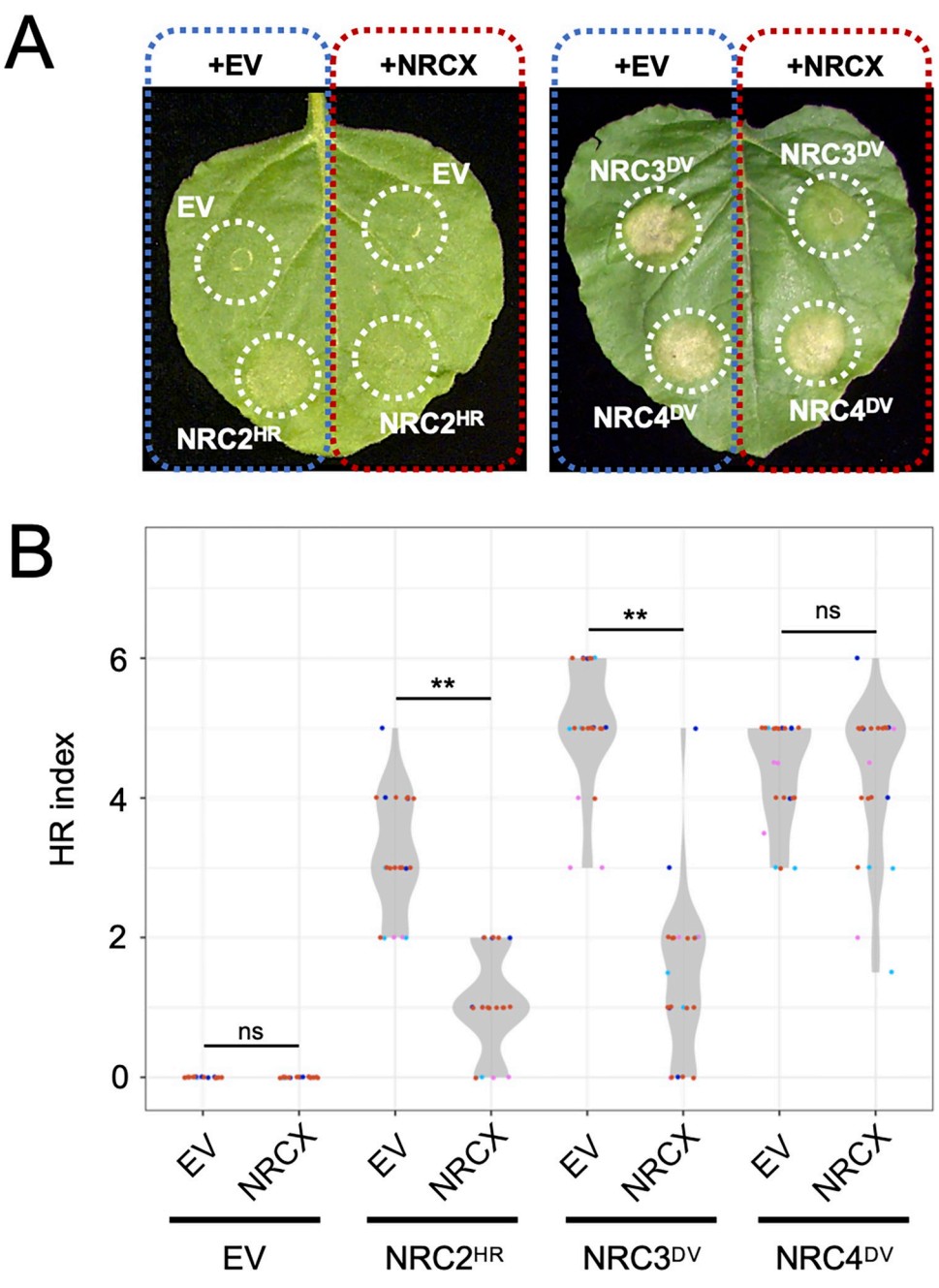

**Fig 7. Overexpression of wild-type NRCX compromises autoactive cell death of NRC2 and NRC3, but not NRC4.**
(**A**) Photo of representative *N. benthamiana* leaves showing autoactive cell death after co-expression of empty vector (EV; control) and wild-type NRCX with NRC2^HR, NRC3^DV and NRC4^DV. Photographs were taken at 5 days after agroinfiltration. (**B**) Violin plots showing cell death intensity scored as an HR index at 5 days after the agroinfiltration. The HR index plots are based on 22 different replicates in four independent experiments. Asterisks indicate statistically significant differences with *t* test (**p<0.01).

*benthamiana* leaves. Overexpression of wild-type SlNRCX compromised autoactive cell death response by NRC3^DV, comparing to the empty vector control (S12 Fig). This result suggests that the modulator function of NRCX in NRC2/NRC3 networks is conserved between *N. benthamiana* and tomato.

### *NRCX* is differentially expressed relative to *NRC2a/b* following activation of pattern-triggered immunity

Our finding that NRCX modulates NRC2 and NRC3 nodes prompted us to investigate the transcriptome dynamics of *NRCX* compared to these *NRC-H* in different plant tissues. To obtain transcriptome data of NRC-H clade members, we extracted total RNA from leaf, root and flower/bud of five-week-old wild-type *N. benthamiana*. Three replicate each from independent plants were subjected to RNA-seq and resulted in 40 million 150-bp paired-end reads per sample. Then, we calculated Transcripts Per Million (TPM) values for *N. benthamiana* *NLR* genes (S5 File). In Fig 8A, we focused on transcriptome profiles of NRC-H with a TPM > 1.0 and found that *NRCX, NRC2a/b/c, NRC3* and *NRC4a/b* genes were expressed in all three tissues, leaf, root and flower/bud. Notably, *NRCX* was about 3 to 5-fold more highly expressed in roots (TPM = 25.5) compared to leaves (TPM = 7.5) and flowers/buds (TPM = 5.2) (Fig 8A). In addition to the *NRC2, NRC3, NRC4* and *NRCX*, we also noted that *NRC5a* and *NRC8a* were expressed in the three tissues, whereas *NRC7a* and *NRC8b* were mainly expressed in *N. benthamiana* roots (Fig 8A). Considering that *NRCX* is coexpressed with *NRC2* and *NRC3* in leaf, root and flower/bud, we propose that NRCX maintains NRC2/3 subnetwork homeostasis during these developmental stages.

Next, we analysed transcriptome data of six-week-old wild-type *N. benthamiana* leaves upon inoculation with bacteria *Pseudomonas fluorescens* 55 [60]. *P. fluorescens* 55 inoculation is considered to trigger PTI in *N. benthamiana*. We, therefore, explored the degree to which *NRC-H* genes are up- or down-regulated upon PTI activation. In total, 61 *NLR* genes were up-regulated, while 14 *NLR* genes were down-regulated in bacterial vs. mock treatment (Fig 8B). Interestingly, the expression of *NRC2a/b* genes was up-regulated in response to *P. fluorescens* 55 inoculation whereas there were no particular differential expression changes with *NRCX* (Fig 8B, S1 Table). Other *NRC-H* genes, such as *NRC2c, NRC3* and *NRC4a/b*, were also unchanged whereas the expression level of *NRC5a/b* increased following bacterial treatment (Fig 8B). We conclude that following activation of immunity in response to bacterial inoculation, *NRC2a/b* become more highly expressed than their paralog and modulator *NRCX*, thereby altering the balance of gene expression between *NRC2a/b* and *NRCX*.

## Discussion

Co-operating plant NLRs are currently categorized into "sensor NLR" for effector recognition or "helper NLR" for immune signalling [54]. These functionally specialized NLR sensors and helpers function in pairs or networks across many species of flowering plants. In this study, we found that NRCX, a recently diverged paralog of the NRC class of helper NLRs, contributes to sustain proper plant growth and is a modulator of the genetically dispersed NRC2/3 subnetwork. We propose that NRCX has evolved to maintain the homeostasis of at least a section of the NRC network (Fig 9). NRCX is also atypical as far as NLR helpers and NRC proteins go, lacking a functional N-terminal MADA motif and the capacity to trigger hypersensitive cell death. At the moment, we cannot categorize NRCX as either a sensor or helper NLR, and it is best described as an NLR modulator.

NLRs are often implicated in spontaneous autoimmune phenotypes in plants and humans [11,61]. In plants, autoimmunity is often observed at the F1 and later generations when genetically distinct plant accessions are crossed [11–13]. One well-studied case of hybrid autoimmunity is induced by a hetero-complex of NLRs from genetically unlinked *NLR* gene loci between DANGEROUS MIX 1 (DM1) from Arabidopsis accession Uk-3 and DM2 from Uk-1 [62–64]. Therefore, hybrid incompatibility can be due to inappropriate activation of mismatched NLRs. Amino acid insertions or substitutions in *NLR* genes can also exhibit autoimmune phenotypes

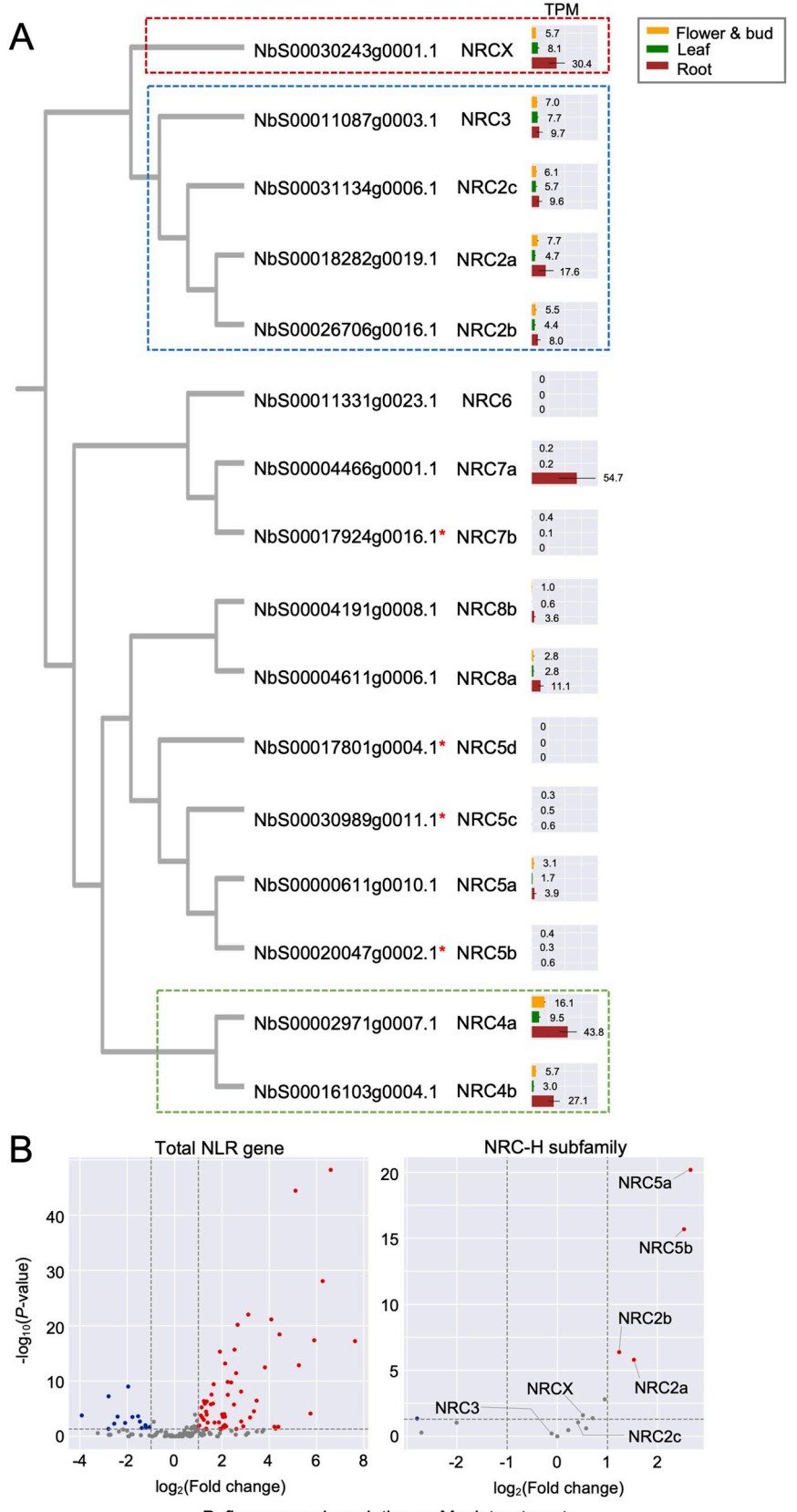

**B**

Total NLR gene

NRC-H subfamily

*P. fluorescens* inoculation vs Mock treatment

**Fig 8. _NRCX_ is differentially expressed relative to _NRC2a/b_ genes in _Nicotiana benthamiana_ leaves after _Pseudomonas fluorescens_ 55 inoculation. (A, B)** TPM values were calculated using RNA-seq data of three different tissues (leaf, root, flower and bud) in 5-weeks old _N. benthamiana_ plants and published transcriptome data of _N. benthamiana_ leaves with mock treatment or _P. fluorescens_ 55 inoculation (Pombo et al., 2019). **(A)** The TPM values analysed from the three different tissue samples are mapped onto phylogeny extracted from the phylogenetic tree in Fig 2C. Red asterisks indicate truncated genes. **(B)** Volcano plots show up-regulated genes (red dots: $\log_2(P.$ _fluorescens_/mock) $\geqq$ 1 and _P_-value $\leqq$ 0.05) and down-regulated genes (blue dots: $\log_2(P.$ _fluorescens_/mock) $\leqq$ -1 and _P_-value $\leqq$ 0.05) in response to _P. fluorescens_ 55 inoculation compared to mock treatment.

in Arabidopsis, such as in the NLR alleles _ssi4_ (G422R), _snc1_ (E552K), _slh1_ (single leucine insertion to RRS1 WRKY domain), _chs1_ (A10T), _chs2_ (S389F) and _chs3-2D_ (C1340Y) [31–34,36,37]. In _chs3-1_, a truncation of the C-terminal LIM-containing domain causes autoimmunity [35]. However, to date, T-DNA insertion or deletion mutants including _chs3-2_, where expression of full-length NLR modules is suppressed, do not show autoimmune phenotypes [32,35,43,65]. Therefore, autoimmune _NLR_ mutants are typically considered gain-of-function mutants. A striking finding in this study is that systemic _NRCX_ silencing resulted in severe dwarfism (Fig 1A and 1B). The dwarf TRV:_NRCX_ plants showed constitutive activation of defense-related genes (Fig 1C and 1D). To our knowledge, this finding is a unique example where VIGS of an _NLR_ gene causes a "lethal" plant phenotype, following the definition by

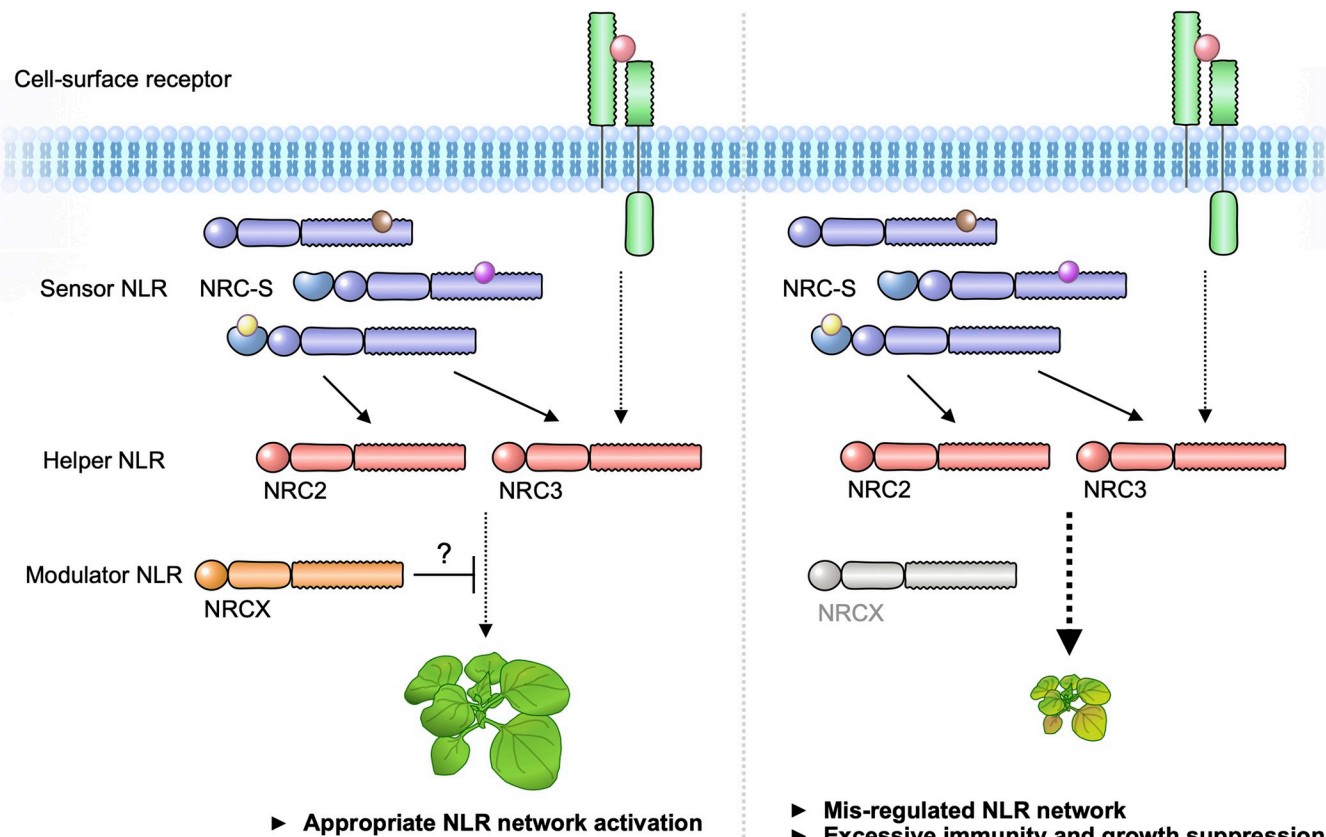

**Fig 9. Modulator NLR has evolved to maintain NLR network homeostasis.** We propose that "Modulator NLR" contributes to NLR immune receptor network homeostasis during plant growth. A modulator NRCX has a similar sequence signature with helper MADA-CC-NLRs, but unlike helpers, NRCX lacks the functional MADA motif to execute cell response. NRCX modulates the NRC2/NRC3 subnetwork composed of multiple sensor NLRs and cell-surface receptor (left). Loss of function of NRCX leads to the enhanced hypersensitive response and dwarfism in _N. benthamiana_ plants (right).

Lloyd et al. [47]. Therefore, *NRCX* is exceptional given that NLR genes tend to act just the opposite way, by causing fitness penalties.

Mutations in *NRC2* and *NRC3* partially suppress the dwarf phenotype of *NRCX*-silenced plants and can be viewed as genetic suppressors of NRCX (Fig 5). These findings inspired us to draw a model which expands our understanding of the NRC network to include NRCX as a modulator of the NRC2 and NRC3 nodes (Fig 9). This network even connects NLRs to cell-surface immune receptors, given that Kourelis et al. [10] recently showed that NRC3 is required for the hypersensitive cell death triggered by the receptor protein Cf-4. We propose that NRCX contributes to the homeostasis of the NRC2/3 subnetwork, which are hubs in an immune network composed of both intracellular NRC-S and the cell-surface receptors. When the expression level of *NRCX* is reduced, NRC2 and NRC3 cause autoimmunity, possibly through inadvertent activation (Fig 9). However, hpRNA-mediated gene silencing of *NRCX* in *N. benthamiana* leaves did not cause autoimmune cell death, although it enhanced the hyper-sensitive cell death elicited by effector recognition (Figs 6 and S9). This indicates that NRCX-mediated dwarfism may be dependent on existence of environmental microbes or viruses and may occur in particular tissues and/or during certain developmental stages. Considering that *nrc2/nrc3* knockout plants do not fully recover from dwarfism (Fig 5), it is possible that *NRCX* silencing leads to a permanent "trigger-happy" status of NRC2/3 and other NLR(s) that ulti-mately perturbs plant growth.

NRCX clusters within the NRC-H subclade, which is populated with proteins with a MADA-CC-NLR architecture (Fig 2). However, the N-terminal MADA motif of NRCX was not functional when swapped into NRC4 (Fig 4), compared to other similar swaps we previ-ously tested [10,24,66]. We conclude that the MADA motif of NRCX has degenerated from the canonical sequence present in the typical NRC helper NLRs and lost the capacity to execute the hypersensitive cell death response. In fact, a polymorphism in the NRCX MADA motif region, Thr-4, is unique to this protein among NRC-H (Fig 2). Moreover, given that functional MADA sequences did not confer cell death activity to NRCX (S7 Fig), NRCX may have become inactive to trigger cell death response by other mutation(s). Therefore, loss of cell death executor activity in NRCX might represent another path in the evolution of networked NLRs besides sensor and helper NLRs, as in the NLR functional specialization "use-it-or-lose-it" model described by Adachi et al. [24] and Kamoun [67]. We propose that NRCX has also functionally specialized into an atypical NLR protein that operates as an NLR modulator.

*N. benthamiana* and tomato NRCX overexpression compromised but did not fully suppress NRC3 autoactive cell death (Figs 7 and S12). Its suppressor activity contrasts with that of the SPRYSEC15 effector from the cyst nematode *Globodera rostochiensis* and AVRcap1b from the oomycete *Phytophthora infestans*, which strongly suppress the cell death inducing activity of autoimmune NRC2 and NRC3 [57]. In the future, it will be interesting to determine the mech-anism by which NRCX modulates the activities of NRC-H proteins, and how that process compares with pathogen suppression of the NRC network.

Our work on NRCX adds to a growing body of knowledge about NLRs that modulate the activities of other NLRs. Classic examples include genetically linked NLR pairs, such as the Pia RGA5/RGA4 pair, in which the sensor RGA5 suppresses the RGA4 autoimmune cell death observed in *N. benthamiana* [68]. In this and other one-to-one paired NLRs, the sensor NLR carries a regulatory role of its genetically linked helper mate that is released by the matching pathogen effector. In contrast, NRCX modulates a genetically dispersed NLR network com-posed of a large number of sensor NLRs and two helper NLRs, NRC2 and NRC3. Recently, Wu et al. [69] showed that overexpression of NRG1C, antagonizes autoimmunity by its para-log NRG1A, *chs3-2D* and *snc1*, without affecting *chs1*, *chs2* autoimmunity and RPS2- and RPS4-mediated immunity. NRG1 is a member of $CC_R$-NLR helper subfamily that triggers cell

death via its N-terminal $CC_R$ domain [70]. Intriguingly, unlike NRCX, NRG1C has a truncated NLR architecture, lacking the whole N-terminal $CC_R$ domain and is therefore unlikely to execute the hypersensitive cell death [69]. The emerging picture is that NRCX and NRG1C are an atypical subclass of helper NLRs, we term modulator NLRs, that lost their cell death executor activity through regressive evolution and evolved to modulate the activities of multiple NLR helper proteins. These examples form another case of functional specialization during the transitions associated with NLR evolution [67]. We need a better appreciation of the diversity of structures and functions that come with the protein we classify as NLR immune receptors and integrate the different ways NLRs contribute to immunity [26].

Plant NLRs are tightly regulated at the transcript level because increased expression of NLRs can result in autoimmune phenotypes and fitness costs [71–74]. However, activation of plant defense is associated with massive up-regulation of *NLR* genes. For instance, in Arabidopsis, dozens of *NLR* genes are up-regulated in response to pathogen-related treatments, such as the bacterial PAMP flg22 [75,76]. The current model is that NLR expression level is maintained at a low basal level but is amplified in the cells upon activation of pathogen-induced immunity. In our analyses, we found that 61 *NLR* genes, including *NRC2a/b*, are up-regulated in *N. benthamiana* leaves following *P. fluorescens* inoculation, while *NRCX* expression levels remain unchanged (Fig 8B). This marked shift in the balance between *NRC2a/b* to *NRCX* expression levels may potentiate and amplify the activity of the NRC network resulting in more robust immune responses (Fig 9).

The NRC superclade forms a large and complex NLR immune receptor network in asterid plants that connects to cell surface receptors [10,53]. Here we show that NRCX modulates the hub NLR proteins NRC2 and NRC3, but doesn't affect their paralog NRC4. Further studies are required to determine the molecular mechanisms underpinning NRCX antagonism of NRC2- and NRC3-mediated immune response and what other mechanisms modulate other sections of the network, such as the NRC4 subnetwork. Our findings also highlight the potential fitness costs associated with expanded NLR networks as the risk of inadvertent activation increases with the network complexity. Further understanding of how NLR network homeostasis is maintained will provide insights for future breeding of vigorous and disease resistant crops.

## Materials and methods

### Plant growth conditions

Wild-type and mutant *Nicotiana benthamiana* were propagated in a glasshouse and, for most experiments, were grown in a controlled growth chamber with temperature 22–25˚C, humidity 45–65% and 16/8 hr light/dark cycle. The NRC knockout lines used have been previously described: *nrc2/3*-209.1.3 and *nrc2/3*-209.3.3 [59], *nrc4*-185.1.2 and *nrc4*-185.9.1 [24], and *nrc2/3/4*-210.4.3 and *nrc2/3/4*-210.5.5 [58].

### Plasmid constructions

The cDNA of *NRCX* was amplified by PCR from *N. benthamiana* cDNA using Phusion High-Fidelity DNA Polymerase (Thermo Fisher) and primers listed in S2 Table. The PCR product was cloned into pICSL01005 (Level 0 acceptor for CDS no stop modules, Addgene no.47996) as a level 0 module for Golden Gate assembly. The cDNA of tomato *NRCX* (*SlNRCX*; Solyc03g005660.3.1) was synthesized via GENEWIZ Standard Gene Synthesis with domesticated mutations on *Bsa*I sites. To generate NRCX-3xFLAG overexpression construct, the pICSL01005::NRCX without its stop codon was used for Golden Gate assembly with pICH51266 [35S promoter+Ω promoter, Addgene no. 50267], pICSL50007 (3xFLAG, Addgene no. 50308) and pICH41432 (octopine synthase terminator, Addgene no. 50343) into

binary vector pICH47732 (Addgene no. 48000). To generate NRCX-4xMyc and SlNRCX-4xMyc overexpression constructs, pICSL01005::NRCX and the synthesized *SlNRCX* were used for Golden Gate assembly with pICSL50010 (4xMyc, Addgene no. #50310) into binary vector pICH86988 (Addgene no. 48076).

To generate virus-induced gene silencing constructs, the silencing fragment (shown in S2 Fig) was amplified from template cDNA of *NRCX* or *Prf* or TRV:NRC2/3, TRV:NRC4, TRV:NRC2/3/4 plasmid [53,77] by Phusion High-Fidelity DNA Polymerase (Thermo Fisher) using primers listed in S2 Table. The purified amplicons were directly used in Golden Gate assembly with pTRV-GG vector according to Duggan et al. [78].

To generate MHD mutants of NRCX, the histidine (H) and aspartic acid (D) residues in the MHD motif were substituted by overlap extension PCR using Phusion High-Fidelity DNA Polymerase (Thermo Fisher). The pICSL01005::NRCX without its stop codon was used as a template. The mutagenesis primers are listed in S2 Table. The mutated NRCX was verified by DNA sequencing of the obtained plasmids.

To generate MADA motif chimera constructs of NRC4 (MADA$^{NRCX}$-NRC4$^{DV}$) and NRCX (MADA$^{NRC2}$-NRCX$^{DV}$, MADA$^{NRC4}$-NRCX$^{DV}$, MADA$^{ZAR1}$-NRCX$^{DV}$, MADA$^{NRC2}$-NRCX$^{HR}$, MADA$^{NRC4}$-NRCX$^{HR}$ and MADA$^{ZAR1}$-NRCX$^{HR}$), we followed a construction procedure of MADA$^{NRC2}$-NRC4$^{DV}$ and MADA$^{ZAR1}$-NRC4$^{DV}$ as described previously [24]. The full-length sequence of *NRC4$^{DV}$*, *NRCX$^{DV}$* and *NRCX$^{HR}$* were amplified by Phusion High-Fidelity DNA Polymerase (Thermo Fisher) using primers listed in S2 Table. Purified amplicons were cloned into pCR8/GW/D-TOPO (Invitrogen) as a level 0 module. The level 0 plasmid was then used for Golden Gate assembly with pICH85281 [mannopine synthase promoter+Ω (MasΩpro), Addgene no. 50272], pICSL50009 (6xHA, Addgene no. 50309) and pICSL60008 [Arabidopsis heat shock protein terminator (HSPter), TSL SynBio] into the binary vector pICH47742 (Addgene no. 48001).

To generate the hpRNA-mediated gene silencing construct, the silencing fragment (shown in S2 Fig) was amplified from NRCX cDNA by Phusion High-Fidelity DNA Polymerase (Thermo Fisher Scientific) using primers listed in S2 Table. The purified amplicon was cloned into the pRNAi-GG vector [79].

Information of other constructs used for the cell death assays were described in S3 Table.

## Virus-induced gene silencing (VIGS)

VIGS was performed in *N. benthamiana* as previously described [80]. Suspensions of Agrobacterium Gv3101 strains carrying TRV RNA1 and TRV RNA2 were mixed in a 1:1 ratio in infiltration buffer (10 mM 2-[N-morpholine]-ethanesulfonic acid [MES]; 10 mM MgCl$_2$; and 150 μM acetosyringone, pH 5.6) to a final OD$_{600}$ of 0.25. Two-week-old *N. benthamiana* plants were infiltrated with the Agrobacterium suspensions for VIGS.

## Bioinformatic and phylogenetic analyses

Based on the NLR annotation and phylogenetic tree previously described in Harant et al. [81], we extracted CC-NLR sequences of tomato, *N. benthamiana*, *A. thaliana* and sugar beet (*Beta vulgaris* ssp. *vulgaris* var. *altissima*). We then added NbS00004191g0008.1 and NbS00004611g0006.1, that lack the p-loop motif in the NB-ARC domain and are therefore missing in the previous CC-NLR list [81], and prepared a CC-NLR dataset (431 protein sequences, S6 File). To newly identify NLR genes from nine Solanaceae species (*N. benthamiana*, *C. annuum*, *C. chinense*, *C. baccatum*, *S. commersonii*, *S. tuberosum*, *S. lycopersicum*, *S. pennellii* and *S. chilense*) and three Convolvulaceae species (*I. triloba*, *I. nil* and *I. trifida*), we ran NLRtracker pipeline [16] to protein databases annotated in each reference genome (S7 File). Amino acid sequences of the annotated NLR genes from the twelve plant species are

listed in S4 File. Amino acid sequences of the NLR datasets were aligned using MAFFT v.7 [82]. The gaps in the alignments were deleted manually and the NB-ARC domain sequences were used for generating phylogenetic tree (S8 and S9 Files). The maximum likelihood tree based on the JTT model was made in RAxML version 8.2.12 [83] and bootstrap values based on 100 iterations were shown in S10 and S11 Files.

NRC-helper proteins were subjected to motif searches using MEME (Multiple EM for Motif Elicitation) v. 5.0.5 [84] with parameters 'zero or one occurrence per sequence, top twenty motifs', to detect consensus motifs conserved in > 80% of input sequences.

## Transient gene expression and cell death assays

Transient gene expression in *N. benthamiana* was performed by agroinfiltration according to methods described by Bos et al. [85]. Briefly, four-week-old *N. benthamiana* plants were infiltrated with Agrobacterium Gv3101 strains carrying the binary expression plasmids. The Agrobacterium suspensions were prepared in infiltration buffer (10 mM MES, 10 mM $MgCl_2$, and 150 μM acetosyringone, pH5.6). To overexpress NRC MADA chimeras and MHD mutants, the concentration of each suspension was adjusted to $OD_{600} = 0.5$. To perform hpRNA-mediated gene silencing experiments in *N. benthamiana* leaves, we infiltrated Agrobacterium strains carrying hpRNA constructs ($OD_{600} = 0.5$), together with different proteins described in S3 Table. Macroscopic cell death phenotypes were scored according to the scale of Segretin et al. [86] modified to range from 0 (no visible necrosis) to 7 (fully confluent necrosis).

To stain dead cells by trypan blue, *N. benthamiana* leaves were transferred to a trypan blue solution (10 mL of lactic acid, 10 mL of glycerol, 10 g of phenol, 10 mL of water, and 10 mg of trypan blue) diluted in ethanol 1:1 and were incubated at 65˚C using a water bath for 1 hour. The leaves were then destained for 48 hours in a chloral hydrate solution (100 g of chloral hydrate, 5 mL of glycerol, and 30 mL of water).

## Protein immunoblotting

Protein samples were prepared from six discs (8 mm diameter) cut out of *N. benthamiana* leaves at 2 days after agroinfiltration and were homogenised in extraction buffer [10% (v/v) glycerol, 25 mM Tris-HCl, pH 7.5, 1 mM EDTA, 150 mM NaCl, 2% (w/v) PVPP, 10 mM DTT, 1x protease inhibitor cocktail (SIGMA), 0.5% (v/v) IGEPAL (SIGMA)]. The supernatant obtained after centrifugation at 12,000 x*g* for 10 min was used for SDS-PAGE. Immunoblotting was performed with HA-probe (F-7) HRP (Santa Cruz Biotech) in a 1:5,000 dilution. Equal loading was checked by taking images of the stained PVDF membranes with Pierce Reversible Protein Stain Kit (#24585, Thermo Fisher Scientific).

## RNA extraction and semi-quantitative RT-PCR

Total RNA was extracted using RNeasy Mini Kit (Qiagen). 500 ng RNA of each sample was subjected to reverse transcription using SuperScript IV Reverse Transcriptase (Thermo Fisher Scientific). Semi-quantitative reverse transcription PCR (RT-PCR) was performed using DreamTaq (Thermo Fisher Scientific) with 25 to 30 amplification cycles followed by electrophoresis with 1.8% (w/v) agarose gel stained with Ethidium bromide. Primers used for RT-PCR are listed in S2 Table.

## RNA-seq analysis

Total RNAs of leaf tissue samples were extracted from four-week-old TRV:*GUS* and TRV:*NRCX benthamiana* plants using RNeasy Mini Kit (Qiagen) or using TRI Reagent (Sigma-

Aldrich) as directed in the protocol. Total RNAs of leaf, root and flower/bud tissue samples were extracted from five-week-old *N. benthamiana* plants using RNeasy Mini Kit (Qiagen). Three replicate each of the samples was sent for Illumina NovaSeq 6000 (40 M paired-end reads per sample, Novogene). Obtained RNA-seq reads were filtered and trimmed using FaQCs [87]. The quality-trimmed reads were mapped to the reference *N. benthamiana* genome (Sol Genomics Network, v0.4.4) using HISAT2 [88]. The number of read alignments in the gene regions were counted using featureCounts [89] and read counts were transformed into a Transcripts Per Million (TPM) value. Differentially expressed genes between TRV:*GUS* and TRV:*NRCX* plants were determined by edgeR through a threshold of $\log_2$FC and false discovery rate ($|\log_2$FC$| > 1$ and FDR $< 0.05$) [90]. For GO analysis, we used GO annotation list (S12 File) extracted from Nicotiana benthamiana v0.4.4 database (Solanaceae Genomics Network) and ran g:GOSt tool in g:Profiler [91] with parameters 'ordered query, only annotated genes, g:SCS threshold, threshold < 0.05' to detect enriched GO terms in differentially expressed gene datasets. Public RNA-seq reads from six-week-old *N. bethamiana* leaves with or without *Pseudomonas fluorescens* 55 inoculation [60], were also analysed as described above (Accession Numbers: SRP118889).

RNA-seq raw reads used for transcriptomic analyses in this study have been deposited under the BioProject accessions: TRV:*GUS*_leaf and TRV:*NRCX*_leaf (PRJEB55392), and five-week old wild-type *N. benthamiana*_leaf, root and flower/bud (PRJEB55516). NRC sequences used in this study can be found in the GenBank/EMBL and Solanaceae Genomics Network (https://solgenomics.net/) databases with the following accession numbers: NbNRC2 (NbS00018282g0019.1 and NbS00026706g0016.1 in *N. benthamiana* draft genome sequence v0.4.4), NbNRC3 (MK692736.1 in GenBank), NbNRC4 (MK692737 in GenBank), NbNRCX (NbS00030243g0001.1 in *N. benthamiana* draft genome sequence v0.4.4) and SlNRCX (Solyc03g005660.3.1 in ITAG3.10).

## Supporting information

**S1 Fig. Phylogenetic tree of NRC superclade.** NRC-sensor (NRC-S) and NRC-helper (NRC-H) proteins identified in Adachi et al. [24] were used for the MAFFT multiple alignment and phylogenetic analyses. The phylogenetic tree was constructed with the NB-ARC domain sequences in MEGA7 by the neighbour-joining method. Each leaf is labelled with different colour ranges indicating plant species, *N. benthamiana* (NbS-), tomato (Solyc-) and sugar beet (Bv-). The NRC-S clade is divided into NLRs that lack an extended N-terminal domain (exNT) prior to their CC domain and those that carry an exNT. Red arrow heads indicate bootstrap support > 0.7. The scale bars indicate the evolutionary distance in amino acid substitution per site.
(TIF)

**S2 Fig. Alignment of *NRCX* and *NRC2b* cDNA sequences.** cDNA sequences of NRCX (NbS00030243g0001.1) and a closely related paralog gene NRC2b (NbS00026706g0016.1) were used for the MAFFT multiple alignment. Red boxes indicate the region used for making virus-induced gene silencing and hairpin RNA constructs of *NRCX*. Search of the *NRCX* 446-bp sequence in SGN VIGS Tool (https://vigs.solgenomics.net/) does not hit off-target candidate genes in Nicotiana benthamiana v0.4.4 and Nicotiana benthamiana v1.0.1 databases.
(TIF)

**S3 Fig. NRC-H subclade shown in Fig 2B.**
(TIF)

**S4 Fig. Phylogenetic tree of NRC-H subclade of nine Solanaceae species.** NRC-helper (NRC-H) proteins identified from *Nicotiana benthamiana*, *Capsicum annuum*, *Capsicum chinense*, *Capsicum baccatum*, *Solanum commersonii*, *Solanum tuberosum*, *Solanum lycopersicum*, *Solanum pennellii* and *Solanum chilense* were used for the MAFFT multiple alignment and phylogenetic analysis. The phylogenetic tree was constructed with the NB-ARC domain sequences in RAxML version 8.2.12 by the maximum likelihood method. NRCX, NRC1/2/3 and NRC4 subclades are labelled with different colour ranges. Red arrow heads indicate bootstrap support > 0.7. The scale bars indicate the evolutionary distance in amino acid substitution per site.
(TIF)

**S5 Fig. Distribution of *NRCX*, *NRC2*, *NRC3* and *NRC4* ortholog genes.** The number of ortholog genes were counted based on NRC-H phylogeny in S4 Fig.
(TIF)

**S6 Fig. 55 independent NRCX MHD mutants do not cause autoactive cell death in *Nicotiana benthamiana*.** Cell death phenotypes were scored at an HR index at 5 days after agroinfiltration to express NRC4$^{WT}$, NRCX$^{WT}$ and the MHD mutants in *N. benthamiana* leaves. Quantification data are from 5 independent biological replicates.
(TIF)

**S7 Fig. The N-terminal 17 amino acids of NRC2, NRC4 and ZAR1 fail to confer cell death activity to NRCX MHD motif mutants.** **(A)** Schematic representation of NRCX MADA motif chimeras. The first 17 amino acid region of NRC2, NRC4 and ZAR1 was swapped into the NRCX MHD motif mutants (NRCX$^{HR}$ and NRCX$^{DV}$), resulting in the NRCX chimeras with MADA sequences originated from other MADA-CC-NLRs. **(B)** Cell death phenotypes induced by NRC4$^{WT}$, NRC4$^{DV}$ and the NRCX chimeras. NRC4$^{WT}$-6xHA, NRC4$^{DV}$-6xHA and the NRCX chimeras were expressed in *N. benthamiana* leaves by agroinfiltration. Photographs were taken at 5 days after the agroinfiltration. **(C)** Violin plots showing cell death intensity scored as an HR index based on 16 or 17 different replicates in three independent experiments. Statistical differences among the samples were analyzed with Tukey's honest significance difference (HSD) test (p<0.01).
(TIF)

**S8 Fig. Co-silencing of *NRC2* and *NRC3* partially suppresses TRV:*NRCX* dwarf phenotype in *Nicotiana benthamiana*.** **(A)** The morphology of 6-week-old *NRCX*-, *NRC2/3/X*-, *NRC4/X*- and *NRC2/3/4/X*-silenced *N. benthamiana* plants. 2-week-old *N. benthamiana* plants were infiltrated with *Agrobacterium* strains carrying VIGS constructs, and photographs were taken 4 weeks after the agroinfiltration. TRV empty vector (TRV:EV) was used as a negative control. **(B, C)** Quantification of the leaf size. One leaf per each plant was harvested from the same position (the 5th leaf from cotyledons) and was used for measuring the leaf diameter. Data was obtained from 25 different VIGS plants in four independent experiments. Statistical differences among the samples were analyzed with Tukey's HSD test (p<0.01). Scale bars = 5 cm. **(D)** Specific gene silencing of *NRCX* or multiple *NRC* genes in TRV:*NRC*-infected plants. Leaf samples were collected for RNA extraction at 3 weeks after agroinfiltration expressing VIGS constructs. The expression of *NRCX* and other *NRC* genes were analyzed in semi-quantitative RT-PCR using specific primer sets. *Elongation factor 1α* (*EF-1α*) was used as an internal control. Scale bars = 5 cm.
(TIF)

**S9 Fig. Hairpin RNA-mediated gene silencing of *NRCX* does not cause cell death in *Nicotiana benthamiana* leaves** (A) Macroscopic cell death phenotype after expressing hpRNA:*GUS*, hpRNA:*NRCX* or *Pto/AvrPto* by agroinfiltration. Photograph was taken at 5 days after the agroinfiltration. **(B)** Cell death was detected by trypan blue staining at 5 days after the agroinfiltration. **(C)** Microscopic cell death phenotype. Dead cells were stained by trypan blue. Images describe representative data of 8 replicates from 2 independent experiments. Scale bars are 300 μm.
(TIF)

**S10 Fig. Time-lapse quantification of NRC-S/AVR-triggered hypersensitive cell death in *NRCX* silenced leaves.** Cell death intensity was scored at 2–5 days after the agroinfiltration as described in Fig 6. Data at 5 days after agroinfiltration is the same with Fig 6B. The HR index plots are based on three independent experiments. Asterisks indicate statistically significant differences with *t* test (*p<0.05 and **p<0.01). Pink and blue line plots indicate mean values of hpRNA:*GUS* and hpRNA:*NRCX* samples at each time point.
(TIF)

**S11 Fig. Time-lapse quantification of NRC-H autoactive cell death in *NRCX* silenced leaves.** Cell death intensity was scored at 2–5 days after the agroinfiltration as described in Fig 6. The HR index plots are based on three independent experiments. Asterisks indicate statistically significant differences with *t* test (**p<0.01). Pink and blue line plots indicate mean values of hpRNA:*GUS* and hpRNA:*NRCX* samples at each time point.
(TIF)

**S12 Fig. Overexpression of wild-type SlNRCX compromises autoactive cell death of NRC3.** **(A)** Photo of representative *N. benthamiana* leaves showing autoactive cell death after co-expression of empty vector (EV; control) and wild-type NRCX with NRC3$^{DV}$. Photographs were taken at 4 days after agroinfiltration. **(B)** Violin plots showing cell death intensity scored as an HR index at 4 days after the agroinfiltration. The HR index plots are based on 27 to 30 different replicates in three independent experiments. Asterisks indicate statistically significant differences with *t* test (**p<0.01).
(TIF)

**S1 Table. Expression ratios of *NRC2*, *NRC3* and *NRC4* compared to *NRCX*.**
(PPTX)

**S2 Table. Primers used in this study.**
(PPTX)

**S3 Table. List of NLR and corresponding AVR effector used in cell death assays.**
(PPTX)

**S1 File. List of up-regulated genes in TRV:*NRCX* leaf compared to TRV:*GUS* control.**
(CSV)

**S2 File. List of down-regulated genes in TRV:*NRCX* leaf compared to TRV:*GUS* control.**
(CSV)

**S3 File. List of genes having GO terms significantly enriched in TRV:*NRCX* compared to TRV:*GUS* control.**
(XLSX)

**S4 File. An NLR dataset including amino acid sequence of 6408 annotated NLRs.**
(FASTA)

**S5 File. Transcriptome profiles of *Nicotiana benthamiana* NLR genes.**
(XLSX)

**S6 File. Amino acid sequences of full-length CC-NLRs used for phylogenetic analysis in Fig 2A.**
(FASTA)

**S7 File. List of reference genome databases used for NLR annotation.**
(XLSX)

**S8 File. Amino acid sequences used for CC-NLR phylogenetic analysis in Fig 2A.**
(FASTA)

**S9 File. Amino acid sequences used for phylogenetic analysis in S4 Fig.**
(FASTA)

**S10 File. CC-NLR phylogenetic tree file in Fig 2A.**
(NWK)

**S11 File. NRC phylogenetic tree file in S4 Fig.**
(NWK)

**S12 File. *Nicotiana benthamiana* GO annotation list used in this study.**
(GMT)

**S13 File. Data and statistical analyses supporting figures and supplemental figures.**
(XLSX)

## Acknowledgments

We are thankful to Joe Win for valuable supports and Jiorgos Kourelis, Lida Derevnina and colleagues for valuable discussions and ideas. We thank Mark Youles of TSL SynBio and photograph office for invaluable technical support.

## Author Contributions

**Conceptualization:** Hiroaki Adachi, Chih-hang Wu, Sophien Kamoun.

**Data curation:** Hiroaki Adachi, Toshiyuki Sakai, Adeline Harant, Hsuan Pai, Kodai Honda, AmirAli Toghani, Jules Claeys, Chih-hang Wu.

**Formal analysis:** Hiroaki Adachi, Toshiyuki Sakai.

**Funding acquisition:** Hiroaki Adachi, Sophien Kamoun.

**Investigation:** Hiroaki Adachi, Toshiyuki Sakai, Chih-hang Wu.

**Methodology:** Hiroaki Adachi, Toshiyuki Sakai, Adeline Harant, Chih-hang Wu.

**Project administration:** Sophien Kamoun.

**Resources:** Hiroaki Adachi, Toshiyuki Sakai, Adeline Harant, Cian Duggan, Tolga O. Bozkurt, Chih-hang Wu.

**Software:** Toshiyuki Sakai.

**Supervision:** Hiroaki Adachi, Sophien Kamoun.

**Validation:** Hiroaki Adachi, Chih-hang Wu.

**Visualization:** Hiroaki Adachi, Toshiyuki Sakai.

**Writing – original draft:** Hiroaki Adachi, Sophien Kamoun.

**Writing – review & editing:** Hiroaki Adachi, Toshiyuki Sakai, Adeline Harant, Hsuan Pai, Cian Duggan, Tolga O. Bozkurt, Chih-hang Wu, Sophien Kamoun.

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
