## [Decision Letter · Decision Letter 0]

7 Jan 2022

Dear Dr Kamoun,

Thank you very much for submitting your Research Article entitled 'An atypical NLR protein modulates the NRC immune receptor network' to PLOS Genetics.

The manuscript was fully evaluated at the editorial level and by independent peer reviewers. The reviewers appreciated the attention to an important problem, but raised some substantial concerns about the current manuscript. Based on the reviews, we will not be able to accept this version of the manuscript, but we would be willing to review a much-revised version. We cannot, of course, promise publication at that time.

If you decide to revise the manuscript for further consideration at PLOS Genetics, please aim to resubmit within the next 60 days, unless it will take extra time to address the concerns of the reviewers, in which case we would appreciate an expected resubmission date by email to plosgenetics@plos.org.

[LINK]

Please do not hesitate to contact us if you have any concerns or questions.

Yours sincerely,

Gitta Coaker, PhD

Associate Editor

PLOS Genetics

Claudia Köhler

Section Editor: Plant Genetics

PLOS Genetics

Editor comments (Gitta Coaker)

All reviewers appreciated the subject matter of the paper and generally felt experiments were well-done, but also suggested several new experiments. These experiments differed between reviewers. Given the general positive comments, I do not think major new experimentation to address all major reviewer comments should be done. Rather, I have provided some guidance below with respect to major comments (minor comments can be textually addressed).

Deposit RNAseq data to a public database

Consider if it is possible to investigate, or at minimum futher discuss, potential reasons for NRCX MADA’s inability to trigger cell death when fused to an autoactive variant, including misfolding.

Reviewer 4 has proposed some informative experiments: “Have the authors tried to see if NRCX or NRCX DV when its alpha 1 helix is swapped with the alpha 1 helix of NRC4 is able to cause cell death? This would give insights whether NRCX would in general be able to function in cell death or whether other constraints are in work. Would it be possible to complement the severe dwarf phenotype of the NRCX silenced plants with a construct that would allow expression of a silencing resistant NRCX version?”

I do not think it is necessary to generate a knockout  in nrcx as it will likely be lethal (but provide more data on specificity - can be in silico data on off targets). Please examine if constitutive defense occurs in the NRCX silenced line. Also, it would be informative to determine if NRCX act as a dominant-negative helper form of NLRs (reviewer 2, also noted in reviewer 3’s comments).

Reviewer's Responses to Questions

**Comments to the Authors:**

Reviewer #1: This manuscript reports identification and characterization of the NLR NRCX that functions as a modulator of the network mediated by the helper NLRs NRC2/3 in Nicotiana benthamiana.

The authors showed that VIGS of NRCX, a paralog of NRC2/3, results in a dwarf phenotype. Then, they showed that MADA motif of NRCX is not functional for cell death-inducing activity. Consistently, mutations in MHD motif of NRCX (that would make NLR autoactive) did not cause cell death in contrast to mutations in other NRCs such as NRC4. Importantly, the dwarf phenotype caused by NRCX VIGS was dependent on NRC2/3. Suppression of NRCX enhanced cell death triggered by autoactive NRC2/3 or NLR activation which depends on NRC2/3. Consistently, overexpression of NRCX inhibited cell death triggered by autoactive NRC2/3. Finally, they showed that expression of other NLRs including NRC2 increased in PTI but not that of NRCX. Based on these results, they propose that NRCX modulates the NRC network.

This study is conceptually similar to a recent publication in Plant Cell (Wu et al in press) where they showed that NRG1c (a truncated NRG1) antagonizes immunity mediated by NRG1a and NRG1b. Nevertheless, there are important differences between two studies. In this study, NRCX and NRC2/3 are genetically unlinked while all NRG1 genes are tandemly aligned on the Arabidopsis genome and NRCX is full length NLR but NRG1c is truncated. Thus, I think that the recent Plant Cell paper does not compromises the significance of this work. An obvious weakness of this study is that the authors did not show the mechanism by which NRCX antagonizes NRC2/3. Another weakness is that the authors used only VIGS but not genetic mutation for NRCX loss of function analysis. However, I believe that these will be an important “future work” in the authors’ laboratory.

The most experiments were well performed and the manuscript is well written although I have one major suggestion and various minor suggestions that the authors might or might not consider for improving this manuscript.

(Major comments)

The incapability of NRCX MADA inducing cell death is interesting considering high amino acid sequence similarities to NRC2 and NRC4 MADA (even higher HMM score than ZAR1 MADA). No cell death triggering activity with the full length (Fig4) might be protein conformation issue but it is surprising to see that even NRCX MADA fusing autoactive NRC4 does not trigger cell death. Can the authors investigate, do AlphaFold2, or discuss more what kind of MADA has cell death inducing activity? In addition, does antagonistic activity of NRCX need NRCX MADA or not? While I think that these would increase the significance and clarity of this study, these may be beyond the scope of this study.

(Minor comments)

1. I think that the authors should deposit the generated raw RNA-seq data to a public database.

2. The authors may combine Figure S4A,C and S8A,B with Figure 5 and 6 and show them as main figures. I find them important.

3. The recent Wu et al Plant Cell paper (http://doi.org/10.1093/plcell/koab285) is not listed in the reference.

4. Page2. “Other Arabidopsis NLRs……” NLRs are not suppressors but mutations are.

5. The last paragraph of Introduction. The authors may make the significance of this study clearer. There are described mechanisms that attenuate NLR networks such as by sensor NLRs, small RNAs… Considering the recent Wu et al Plant Cell, “unknown” may be too much.

6. The first two sentences of Results seem redundant with Introduction and may not necessary.

7. Page 4. “NRCX silencing could lead to NLR mis-regulation, thereby resulting in the dwarf phenotype in N. benthamiana plants.” This sentence may confuse readers and may be removed.

8. Page 10. “Our finding that NRC2 and NRC3 are genetic suppressors of NRCX raises the possibility……” Mutations in NRC2 and NRC3 are genetic suppressors of phenotypes caused by NRCX silencing.

9. Page 12. “….. all three tissues, leaf, root and flower/bud at relatively similar ratios.” I would not call these “similar ratios”.

10. Page 12. “These transcriptome profiles suggest….” I feel this conclusion is not well supported by Fig7.

11. Page 12. “We conclude that following activation of…..” The authors may rewrite this.

12. Page 14. “To our knowledge, this finding is a unique example where loss-of-function of….” As the authors did not investigate complete loss of function of NRCX, the authors should reconsider this conclusion. The authors only showed results of NRCX VIGS, raising a slight concern that phenotypes caused by NRCX VIGS and NRCX knockout might not be the same.

13. Page 14. “NRC2 and NRC3 partially suppress the dwarf phenotype….” Mutations in NRC2 and NRC3 do.

14. Page 14. “This indicates that NRCX-mediated dwarfism is trigger-dependent……… other NLR(s) that ultimately perturbs plant growth.” The authors may explain this in more detail as readers might wonder what “the trigger” is. Can it be the virus used in VIGS? Or some environmental microbes? I do not understand “similar to pathogen recognition”.

Reviewer #2: This manuscript by Adachi et al. describes their functional analysis of an atypical Solanaceous helper NRC NRCX. In contrast to other NRCs, NRCX lacks a functional N-terminal MADA motif and the ability to trigger HR. Interestingly, silencing of NRCX in Nicotiana benthamiana lead to a dwarf phenotype, which is partially dependent on NRC2/3 but not NRC4. Moreover, RNA interference of NRCX enhances while overexpression of NRCX compromises NRC2 and NRC3-dependent cell death in N. benthamiana. These data suggest a negative role of NRCX in NRC2/3-mediated immune pathways/networks. Although the genetic data are largely convincing, biochemical supports are lacking. Here are some concerns the authors should address during revision.

1. Fig 1. What is the evidence supporting that silencing of NRCX specifically causes the dwarf phenotype? Due to the various problems associated with vigs, the most reliable approach for corroboration is to generate knockout alleles of nrcx for phenotypic examination (the ko mutant is predicted to show the same or worse dwarfism than the vigs line). There is no description in the current manuscript on how many vigs lines were examined, and how reproducible the phenotype is. Does the vigs line affect the transcription or protein levels of NRC2/3 (Fig S4D is not quantitative enough to address mild transcriptional differences)? Answers to these questions may help explain how NRGX negatively regulate NRC2/3.

2. There is very limited analysis of the dwarf phenotype in the NRCX silencing line. Is it developmental or due to constitutive defense activation? What are its immune phenotypes (tested with pathogens)?

3. Related to #2, how about the immunity related phenotypic analysis of the NRCX overexpression lines? Are they like the nrc2/3 knockout lines?

4. From data in Fig 2-3, it seems that NRCX probably forms a non-functional N-terminal α1 helix and this may affect the activities of autoactive NRC2/3. It is possible that NRCX may serve as a dominant-negative form of helper NRCs, reminiscent of NRG1C reported recently. Thus, some simple protein-protein interaction experiments should be conducted to examine the interactions between NRCX and NRC2/3/4. The authors can test if NRCX competes with NRC2/3 to oligomerize, which may result in the formation of non-functional NRC resistosomes.

5. In the abstract, the authors claim that NRCX has evolved to contribute to the homeostasis of the genetically unlinked NLR network. However, how it evolved is not clearly described in the manuscript. Is it co-existing with NRC2/3, and to what extent? From the phylogenetic tree, it seems that NRCX forms a small clade with a tomato NRCX homolog. To corroborate their conclusions on NRCX, the function of SlNRCX should be examined to see if it also works as a negative regulator.

Minor points:

1. Page 3, explain what is CCR-NLRs before using the abbreviation.

2. Page5, the authors omitted an ‘it’ in the last paragraph. It should be ‘despite its sequence conservation, it probably forms a non-functional N-terminal α1 helix’.

3. Page 8, change ‘HR and DV’ to ‘H474R and D475V’.

4. Change ‘Nicotiana benthamiana’ to ‘N. benthamiana’ in the titles of each section.

5. Page 12, to avoid confusion, the authors should use the bacterial strain name Pseudomonas fluorescens 55, which is virulent on N. benthamiana.

Reviewer #3: The review has not been uploaded as an attachment.

In their report, Adachi and coworkers describe the identification of a new paralogous NRC helper NLR in Nicotiana benthamiana and show functional analysis supporting its role as an attenuator of NLR-mediated immunity in solanaceous plants. Their observations are presented in the context of hybrid necrosis, NLRs’ phylogeny and evolution, tuning of ETI in plants and nicely couple with authors’ earlier work on NLR networking and their functional redundancy. The paper is clear and well written.

A few comments for author/editor consideration to further improve this production:

1. There is no information provided on possible off-target effects of silencing constructs. How unique were the sequences selected to trigger silencing? Were they the same for VIGS and dsRNA constructs?

2. In the model presented in Fig 8 the authors place NRCX downstream of NRC2&3 activation but the fact that no necroses were present after in-spot NRCX silencing may imply that NRCX associates with other NRCs prior to their activation; in other words: transient, hpRNA-mediated silencing might have not affected the complexes existing in the cell and it was not enough time for NRC2&3 to accumulate to levels triggering defenses. Is it possible that NRCX acts in fact upstream of other NRCs or competing with them for interactions w/ NRC-Ss? The autoimmunity of NRCs may be their “own” properties but it may also be a result of “oversensitive” sensors activating them in the absence of pathogen.

3. The term of “RNA interference” (RNAi) was originally introduced in animals and indeed is sometimes used in plants interchangeably with post-transcriptional gene silencing but because two different techniques of gene silencing have been used (VIGS involves RNA interference too) I would abstain from using this term here and use “hairpin RNA-mediated Postranscriptional Gene Silencing” or “hpRNA-PTGS” instead. To emphasize the difference from VIGS (that affects the whole plant) and hpRNA-PTGS, the authors may also consider using simply “in spot” or ATTA-triggered silencing – just a suggestion. Note, that RNAi was also used in the Discussion.

4. Minor: a few more reports involving crop plants may be added to the discussion where NLR-bearing loci are responsible for incompatibility and/or hybrid necrosis, for example the work involving wheat from B. Keller’s lab.

Reviewer #4: Review of PGENETICS-D-21-01645 “An atypical NLR protein modulates the NRC immune receptor network” by Adachi et al.

In this manuscript the authors provide evidence for the existence of a atypical CC-NLR of the N. benthamiana NRC class that negatively regulates the NRC immune network. The topic is nicely introduced and the introduction provides the reader with sufficient information to follow the conclusions. The experiments are very well performed and of high quality. The figures are very good and present the relevant data to follow the authors conclusions. The discussion is putting the results into the context of the current knowledge of NLR networks and enables the reader to further think about the issue tackled in the current manuscript.

The authors use VIGS in N. benthamiana to show that NRCX silencing induces a dwarf autoimmune related phenotype that is partially dependent on the presence of wildtype NRC2 and 3. They further show that changing NRCX expression levels affects NRC2 and 3, but not NRC4, function in sensor NRC-triggered immunity. The authors further suggest by swap-experiments between NRCX and NRC4 and by mutational analysis of NRCX that NRCX may present an atypical NLR (NRC NLR) that lacks cell death activity per se and may have evolved to regulate the NRC network. I really like this work and the manuscript, but I do have some suggestions to improve the claims and the manuscript.

Overall, I do think it is a nice piece of work out of the very productive Kamoun lab and provides great new insights into how NLRs are regulated and how they execute immunity.

below are some of my concerns and suggestion.

main issues/questions:

abstract: An NRCX ‘sequence’-ortholog is not found in all Solanaceae species, right? Thus, it would suggest that NRCX has not evolved as an essential NLR NLR (NRC) gene in all Solanaceae. I suggest to specify this already in the abstract and to be clearer about this – NRCX regulates the NRC network in N. benthamiana, there might be other NLRs, not sequence ortholog to NbNRCX, regulating this network or there may be other regulatory mechanisms. What is the authors opinion on this?

The experiments showing that NRCX and its alpha1 helix are unable to induce HR, even if swapped onto NRC4 DV is convincing (Fig.3). However, the follow up experiment testing whether NRCX MHD mutants can cause HR is (Fig. 4), at least to me, not the most logical one. I would have guessed that NRCX DV is unable to cause HR, because its N-terminal alpha helix is not (cell death) functional. Have the authors tried to see if NRCX or NRCX DV when its alpha 1 helix is swapped with the alpha 1 helix of NRC4 is able to cause cell death? This would give insights whether NRCX would in general be able to function in cell death or whether other constraints are in work.

Would it be possible to complement the severe dwarf phenotype of the NRCX silenced plants with a construct that would allow expression of a silencing resistant NRCX version? If yes, this would also make it possible to test whether NRCX P-loop function is required for the regulation or one could even test how a ‘auto-activation’ (MHD mutant version mimics potentially ATP binding oligomerization) of NRCX affects this regulation. This approach could be also used in the RNAi experiments, right? I think this would at least add up to the potential mechanism of how NRCX regulates NRC2/3.

Is a NRCX knock out plant viable? If not, can such a plant be generated in a nrc2/3 mutant background?

Did the authors get RNAseq data for NRCX silenced plant tissue? Are the expression levels of NRC2, 3 or any other NRCs changed in such a plant?

minor issues

abstract: please explain the NRC abbreviation as you did for NLR proteins.

page 2 introduction: The sentence listing Arabidopsis NLR mutants is not correct – at least in my understanding. Should it not say: “Mutations in other Arabidopsis NLRs, such as laz5, adr1 ...are genetic suppressors of autoimmunity or cell death phenotypes.”? Because the phenotype depends on the wildtype genes and thus these cannot be genetic suppressors!

Figure 2: Please indicate what the red asterisks mean close to the labels of the truncated NLRs.

Figure 4. According to my comment above in the main issue section I would suggest to move figure 4 to the supplement.

Results section for figure 4: I would disagree that this results clearly indicate that NRCX does not have the capacity to induce cell death. The alpha 1 helix of NRCX is insufficient and thus most likely the whole protein does not induce cell death, but I suppose if the alpha 1 helix would be swapped with the one of NRC4 this chimeric protein would be able to induce cell death – at least this should be tested to make this claim.

last paragraph of results: The authors write that P. fluorescence is an avirulent pathogen on N. benthamiana and that it triggers PTI. However, according to the general terminology an avirulent pathogen is considered to be detected by the plants NLR system and trigger ETI. Do the authors meant a non-virulent (not or only weakly growing/infecting) pathogen? Please explain.

discussion: I would suggest to tone down the claim that NRCX is an essential plant gene, since the authors show it only for N. benthamiana and not for any other NRCX containing species.

**Have all data underlying the figures and results presented in the manuscript been provided?**

Reviewer #1: **No: **

Reviewer #2: Yes

Reviewer #3: Yes

Reviewer #4: None

PLOS authors have the option to publish the peer review history of their article (what does this mean?). If published, this will include your full peer review and any attached files.

Reviewer #1: No

Reviewer #2: No

Reviewer #3: **Yes: **Tadeusz Wroblewski

Reviewer #4: No

---

## [Decision Letter · Decision Letter 1]

27 Oct 2022

Dear Dr Kamoun,

We are pleased to inform you that your manuscript entitled "An atypical NLR protein modulates the NRC immune receptor network in *Nicotiana benthamiana*" has been editorially accepted for publication in PLOS Genetics. Congratulations!

Yours sincerely,

Gitta Coaker, PhD

Academic Editor

PLOS Genetics

Claudia Köhler

Section Editor

PLOS Genetics

Comments from the reviewers (if applicable):

Reviewer's Responses to Questions

**Comments to the Authors:**

Reviewer #2: The authors have carefully addressed most of my concerns, thank you for the effort.

To highlight the importance of this work, I would suggest the authors to add some more discussion to compare the potential mechanisms of NRCX with NRG1C to clarify their similarities and differences (in terms of genomic arrangements, specificities, potential biochemical mechanisms, etc.).

Reviewer #4: The authors responded to all my comments and included new experiments to support their findings. I am fully satisfied with the current version of the MS. I would also like to congratulate the authors for this nice piece of work

**Have all data underlying the figures and results presented in the manuscript been provided?**

Reviewer #2: Yes

Reviewer #4: Yes

PLOS authors have the option to publish the peer review history of their article (what does this mean?). If published, this will include your full peer review and any attached files.

Reviewer #2: No

Reviewer #4: No

**Data Deposition**

http://datadryad.org/submit?journalID=pgenetics&manu=PGENETICS-D-21-01645R1

**Press Queries**

---

## [Editor Report · Acceptance letter]

8 Dec 2022

PGENETICS-D-21-01645R1 

An atypical NLR protein modulates the NRC immune receptor network in *Nicotiana benthamiana*

Dear Dr Kamoun, 

We are pleased to inform you that your manuscript entitled "An atypical NLR protein modulates the NRC immune receptor network in *Nicotiana benthamiana*" has been formally accepted for publication in PLOS Genetics! Your manuscript is now with our production department and you will be notified of the publication date in due course.

With kind regards,

Zsofia Freund

PLOS Genetics

On behalf of:
